# Explainable HCC Diagnosis on Dynamic Contrast-Enhanced MRI with a Li-RADS Concept Bottleneck

**Killian Monnin**[1,2]                               KILLIAN.MONNIN@UNIL.CH

**Patrick Jeltsch**[2]                                PATRICK.JELTSCH@CHUV.CH

**Lucia Fernandes-Mendes**[2]                         LUCIA.FERNANDES-MENDES@CHUV.CH

**Vasco Cazzagon**[2]                                 VASCO.CAZZAGON@CHUV.CH

**Murat Yüce**[3]                                     MURAT.YUCE@MSSM.EDU

**Vivek Yadav**[3]                                    VIVEK.YADAV@MSSM.EDU

**Mario Jreige**[2]                                   MARIO.JREIGE@CHUV.CH

**Marianna Gulizia**[2]                               MARIANNA.GULIZIA@CHUV.CH

**Montserrat Fraga Christinet**[2]                    MONTSERRAT.FRAGA@CHUV.CH

**Raphaël Girardet**[4]                               RAPHAEL.GIRARDET@HEALTH.WA.GOV.AU

**Clarisse Dromain**[2]                               CLARISSE.DROMAIN@CHUV.CH

**Bachir Taouli**[3]                                  BACHIR.TAOULI@MOUNTSINAI.ORG

**Naïk Vietti-Violi**[1,2]                            NAIK.VIETTI-VIOLI@CHUV.CH

**Jonas Richiardi**[1,2]                              JONAS.RICHIARDI@CHUV.CH

[1] *Department of Radiology, Lausanne University Hospital, Lausanne, Switzerland*

[2] *University of Lausanne, Lausanne, Switzerland*

[3] *Biomedical Engineering and Imaging Institute, Icahn School of Medicine at Mount Sinai, New York, USA*

[4] *Department of Radiology, South Metropolitan Health Service, Murdoch, Australia*

**Editors:** Accepted for publication at MIDL 2026

## Abstract

We propose an explainable end-to-end framework for hepatocellular carcinoma (HCC) diagnosis on dynamic contrast-enhanced (DCE) liver MRI. Our method embeds Liver Imaging Reporting and Data System (Li-RADS)–inspired concepts into the network via a multihead concept bottleneck. A 2.5D EfficientNet backbone processes lesion-centred multiphase MRI crops, and a 4-head architecture jointly predicts continuous soft labels for non-rim arterial phase hyperenhancement (APHE), portal venous/delayed washout and capsule, lesion morphology, and a LR-5 score (definite HCC vs non-HCC) based on the Li-RADS guidelines. Soft labels are derived automatically from intra-lesional, peri-lesional and parenchymal intensity patterns, and the network is trained with uncertainty-weighted losses to balance concept prediction, contrast regression and HCC classification. On our cohort, the Li-RADS–inspired bottleneck substantially improves NormGrad explanation accuracy, geometric stability and intensity robustness while maintaining PR AUC comparable to a single-head baseline, highlighting an interpretable alternative to a black-box HCC classifier.

**Keywords:** HCC, Classification, Explainable AI, MRI, Concept bottleneck model

## 1. Introduction

Hepatocellular carcinoma (HCC) is a major cause of cancer mortality (Rumgay et al., 2022). Early screening and diagnosis are an important way to reduce mortality, as are improved imaging techniques such as dynamic imaging. While this can improve diagnostic accuracy, it also introduces extra burden because several images must be read by radiologists. In this work, we therefore focus on automated lesion characterization on dynamic contrast–enhanced MRI (DCE-MRI). HCC is the only cancer that can be diagnosed on imaging - based on Liver Imaging Reporting and Data System (Li-RADS) score and the presence of cirrhosis, without the need for biopsy. In clinical routine, non-invasive diagnosis of HCC thus relies on Li-RADS score, which formalizes a phase-ordered vascular pattern: non-rim arterial-phase hyperenhancement (APHE) followed by washout or capsule on portal venous/delayed phases—as major features for HCC diagnosis. This pattern reflects contrast kinetics across arterial, portal-venous, and delayed phases, making DCE-MRI a natural substrate for lesion characterization (Chernyak et al., 2018; Shin et al., 2021).

Recent deep learning studies report strong HCC vs non-HCC discrimination on multiphasic MRI, including gadoxetic-enhanced protocols and multicenter cohorts, yet typically treat phases as stacked channels or late-fused features without explicitly encoding temporal order (Sarfati et al., 2025; Li et al., 2025; Wang et al., 2023). Parallel works explore Li-RADS assistance or end-to-end category prediction, but supervision often targets labels (LR-3/4/5/M, APHE/washout present/absent) rather than the ordered APHE→washout chronology that defines the criteria (Sarfati et al., 2025; Li et al., 2025; Wang et al., 2023; Stollmayer et al., 2025). In terms of reliability, uncertainty methods (e.g., Monte-Carlo dropout, post-hoc calibration) have been adopted in medical imaging, and explainability techniques (e.g., NormGrad) are widely used in radiology; however, these approaches audit decisions after training and do not constrain which phase should carry signal nor how uncertainty should reflect temporal evidence, in contrast to concept bottleneck models that explicitly intervene at the concept level (Koh et al., 2020; Rebuffi et al., 2019; Nair et al., 2020; Gal and Ghahramani, 2015).

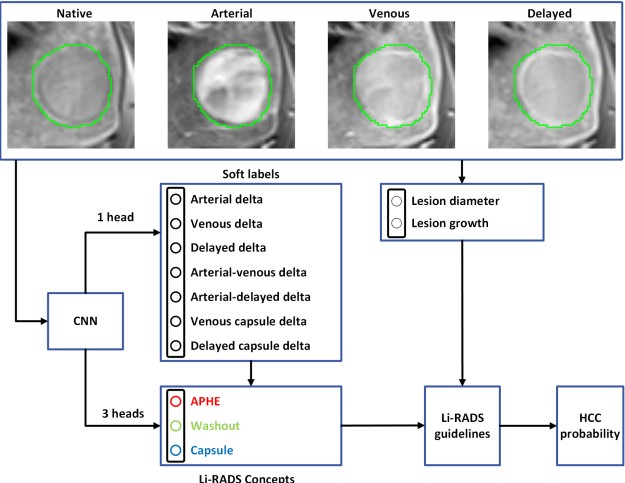

Figure 1: Bottleneck concept model for LR-5 prediction

We address this gap with an end-to-end Li-RADS–inspired concept bottleneck framework (Figure 1), predicting APHE, washout and capsule as intermediate concepts, which are then combined with lesion morphology to predict an LR-5 score through a soft implementation of Li-RADS guidelines.

Our main contributions are:

- **Clinically grounded system.** We introduce a Li-RADS–aligned diagnostic support system for DCE-MRI that outputs the final LR-5 decision together with radiologist-relevant intermediate cues (e.g., APHE, washout, capsule), enabling verification of the model's reasoning in clinically meaningful terms (Chernyak et al., 2018).

- **Method components.** Building on prior work, we leverage concept bottleneck models (including test-time concept interventions) and NormGrad-style saliency for localization (Koh et al., 2020; Rebuffi et al., 2019; Raatikainen and Rahtu, 2025). We tailor these components to the multiphase Li-RADS setting by introducing:

  - DCE MRI/contrast-driven continuous soft labels for major features.
  - A differentiable implementation of the Li-RADS decision rules for HCC diagnosis on DCE-MRI.
  - A Li-RADS-specific robustness/stability and intervention evaluation protocol that quantifies how concept corrections propagate to the final LR-5 prediction.

We address three questions: (i) does the Li-RADS–inspired concept bottleneck improve NormGrad faithfulness and stability compared to a single-head baseline? (ii) do physics-driven soft labels improve concept prediction and explanation robustness, and at what cost in LR-5 performance? (iii) how much LR-5 performance can be recovered through Li-RADS concept interventions without retraining?

## 2. Materials and Methods

### 2.1. Study design and cohorts

This retrospective, single-center study included two cohorts of patients at high risk of HCC, each imaged with extracellular contrast-enhanced liver MRI. Institutional review board approval was obtained, and the requirement for informed consent complied with local regulations. The Surveillance cohort comprised 101 patients with chronic liver disease referred for liver MRI (Girardet et al., 2023). The Pre-Ablation cohort originated from a prospective registry and included 67 patients undergoing MRI prior to percutaneous ablation (Vietti Violi et al., 2018). The two cohorts were merged to obtain a dataset of 250 lesions, including 181 HCC, after excluding cases with advanced HCC, missing phases or annotations, poor-quality images, or registration failure. Lesion size (maximum diameter) ranged from 6–104 mm (median: 20 mm; mean±SD: $23.57 \pm 13.62$ mm), with 8 (3.2%) lesions <10 mm, 115(46.0%) measuring 10–19 mm, and 127(50.8%) $\geq$20 mm; HCC lesions were larger on average than non-HCC lesions ($26.46 \pm 14.58$ mm vs. $15.99 \pm 5.92$ mm).

We additionally evaluated our method on the public LiverHccSeg dataset, derived from TCGA-LIHC on TCIA. It contains one multiphasic contrast-enhanced T1-weighted MRI

study per patient (pre-contrast, arterial, portal-venous, and delayed phases), with co-registered phases and manual liver/tumor segmentations provided by two board-certified abdominal radiologists (17 cases for liver masks; 14 patients / 16 HCC lesions for tumor masks). A consensus scientific reading provides LI-RADS-based labels, including major imaging features such as APHE, washout, and capsule, which we use for external evaluation. (Gross et al., 2023). To better match our target setting, we excluded HCCSeg cases with very large lesions ($> 70$mm of diameter) from this external evaluation. Lesion size (maximum diameter) had a median of 40.3 mm (mean±SD: 41.5±17.8 mm); 0/10 lesions were <10 mm, 1/10 (10%) measured 10–19 mm, and 9/10 (90%) were ≥20 mm.

## 2.2. Data splits and evaluation

The data were split with 3-fold cross-validation patient-wise and stratified by HCC status. A held-out test set (27 patients, 55 lesions, 38 HCC) was reserved for testing. All splits were patient-disjoint across training, validation, and testing. We report area under the receiver-operating characteristic curve (ROC AUC), area under the precision-recall curve (PR AUC), and Matthews Correlation Coefficient (MCC).

## 2.3. Image Parameters and Analysis

We used multiphasic contrast-enhanced T1-weighted liver MRI comprising a native pre-contrast acquisition (no intravenous contrast) and three post-contrast dynamic phases acquired after intravenous administration of an extracellular gadolinium agent: late arterial (20 s after injection), portal venous (3 min), and delayed (5 min). These exams were acquired on 1.5T/3T scanners according to standard clinical protocols for HCC surveillance and pre-ablation. Full MRI acquisition parameters and the annotation protocol are provided in Appendix A.

## 2.4. Image pre-processing and 2.5D lesion crops

The liver mask used for preprocessing and ROI definition was generated automatically using a pre-trained nnU-Net from our previous work (Monnin et al., 2025). For each patient, all multiphase liver MRI volumes were normalised by dividing each phase by the median signal of the erector spinae muscles. During training and validation, lesion-centred liver volumes were pre-processed with a MONAI-based pipeline: images and masks were converted to channel-first RAS orientation, resampled to an in-plane resolution of $224 \times 224$, and augmented with random flips, 90° rotations and small affine transforms applied jointly to image, liver and lesion masks. To reduce shortcut learning on extra-hepatic background, we applied a background suppression transform that attenuates voxels outside the union of liver and lesion masks (details in Appendix B), followed by intensity normalisation of non-zero voxels.

For each annotated lesion, we extracted a 3D lesion-centred crop, identified all axial slices with non-zero segmentation and constructed a 2.5D multi-phase input by selecting (or symmetrically replicating) five consecutive slices around the lesion centre across the four phases (native, arterial, portal venous, delayed). The classifier input consisted of the 4 multiphasic image channels concatenated with the binary lesion mask as an additional

channel. In addition, for the computation of Li-RADS–inspired soft labels only, we generated auxiliary phase-difference images (dynamic phase minus native, per-lesion min–max normalised and clipped to $[0, 1]$), which were used solely for soft-label computation and not as network inputs. More details are provided in Appendix C.

## 2.5. Li-RADS Concepts and Soft Labels

### 2.5.1. Li-RADS–inspired soft labels

Based on the phase-difference images (dynamic phases minus native) and the corresponding liver and lesion masks, we define lesion-wise continuous soft labels that approximate Li-RADS major features.

For each connected lesion component $K_i$ inside the liver mask $L$, we construct a dilated region $K_i^{\mathrm{dil}}$ by morphological expansion (Fig. 2), and derive the peri-lesional ring and surrounding parenchyma as $R_i = K_i^{\mathrm{dil}} - K_i$ and $P_i = L - K_i^{\mathrm{dil}}$

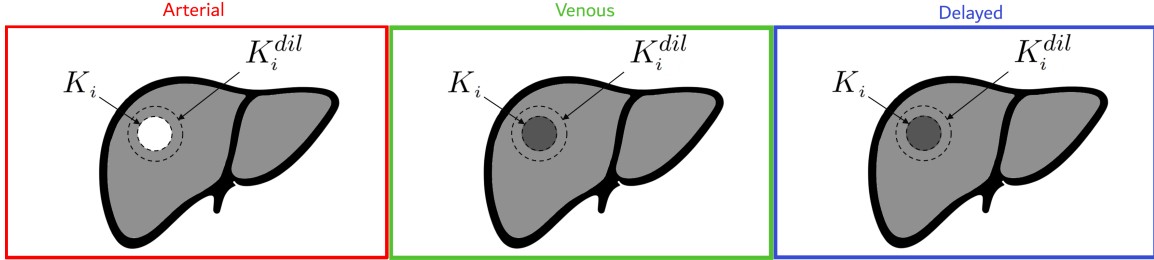

Figure 2: Regions used to compute phase-specific soft labels for a lesion component $K_i$: lesion $K_i$, dilated lesion $K_i^{\mathrm{dil}}$, peri-lesional ring $R_i$ and parenchyma $P_i$ inside the liver $L$.

Let $I_p(x)$ denote the phase-difference image at phase $p \in \{1, 2, 3\}$ (arterial, venous, delayed) and $K_{p,i}^{\mathrm{in}} \subseteq K_i$ the phase-specific subset of lesion voxels used at phase $p$. We compute robust median intensities

$$m_{p,i}^{\mathrm{in}} = \mathrm{median}\{I_p(x) : x \in K_{p,i}^{\mathrm{in}}\}, \quad m_{p,i}^{\mathrm{ring}} = \mathrm{median}\{I_p(x) : x \in R_i\},$$
$$m_{p,i}^{\mathrm{par}} = \mathrm{median}\{I_p(x) : x \in P_i\}.$$

Phase-wise lesion–parenchyma deltas summarise relative hyper-/hypo-enhancement,

$$d_{p,i}^{\mathrm{par}} = m_{p,i}^{\mathrm{in}} - m_{p,i}^{\mathrm{par}}, \qquad p = 1, 2, 3,$$

while additional deltas capture washout and peri-lesional rim enhancement:

$$d_{p,i}^{\mathrm{wash}} = d_{p,i}^{\mathrm{par}} - d_{1,i}^{\mathrm{par}}, \qquad d_{p,i}^{\mathrm{ring}} = m_{p,i}^{\mathrm{in}} - m_{p,i}^{\mathrm{ring}}, \qquad p = 2, 3.$$

Collecting these seven quantities yields a soft-label vector

$$\mathbf{s}_i = \left(d_{1,i}^{\mathrm{par}}, d_{2,i}^{\mathrm{par}}, d_{3,i}^{\mathrm{par}}, d_{2,i}^{\mathrm{wash}}, d_{3,i}^{\mathrm{wash}}, d_{2,i}^{\mathrm{ring}}, d_{3,i}^{\mathrm{ring}}\right),$$

used as regression target for the contrast head. A simple mapping $(a_i, w_i, c_i) = f(\mathbf{s}_i)$ then produces soft Li-RADS concept scores for APHE ($a$), washout ($w$) and capsule ($c$), which enter the multi-task loss; more details are provided in Appendix D.

### 2.5.2. MULTI-HEAD CONCEPT PREDICTION

To leverage complementarities between Li-RADS concepts and to model them explicitly, we train our concept bottleneck models in a multi-task setting. For each lesion $i \in \{1, \ldots, N\}$, a first regression head predicts the soft-label vector

$$\hat{\mathbf{s}}_i = \left( \hat{d}_{1,i}^{\mathrm{par}}, \hat{d}_{2,i}^{\mathrm{par}}, \hat{d}_{3,i}^{\mathrm{par}}, \hat{d}_{2,i}^{\mathrm{wash}}, \hat{d}_{3,i}^{\mathrm{wash}}, \hat{d}_{2,i}^{\mathrm{ring}}, \hat{d}_{3,i}^{\mathrm{ring}} \right)$$

On top of the shared encoder, three concept heads output logits for APHE, washout and capsule:

$$\mathbf{z}_i = \left( z_i^a, z_i^w, z_i^c \right), \qquad \mathbf{y}_i = \left( y_i^a, y_i^w, y_i^c \right),$$

where $a, w, c$ denote APHE, washout and capsule. We define

$$p_i^k = \sigma \left( z_i^k \right), \qquad k \in \{a, w, c\},$$

with $\sigma$ the logistic sigmoid, so that $p_i^a, p_i^w, p_i^c$ are the model-predicted probabilities of the three Li-RADS major features. The LR-5 probability $p_i^\ell$ is then obtained by feeding these concept probabilities into the differentiable Li-RADS rule described in Section 2.5.3.

### 2.5.3. DIFFERENTIABLE IMPLEMENTATION OF LI-RADS

In Li-RADS v2018 (Figure 3), a lesion can be categorized as LR-5 (definite HCC) based on its diameter and the presence of major imaging features in at-risk patients. In our setting, we focus on the two main size-dependent cases: (i) nodules with non-rim APHE and a diameter between 10 and 19 mm, where LR-5 requires APHE together with either washout or threshold growth (TG), and (ii) nodules with non-rim APHE and a diameter $\geq 20$ mm, where LR-5 requires APHE together with at least one major feature among washout, enhancing capsule, or TG.

**CT/MRI Diagnostic Table**

| Arterial phase hyperenhancement (APHE) | | No APHE | | Nonrim APHE | | |
|---|---|---|---|---|---|---|
| Observation size (mm) | | < 20 | ≥ 20 | < 10 | 10-19 | ≥ 20 |
| Count additional major features: | None | LR-3 | LR-3 | LR-3 | LR-3 | LR-4 |
| • Enhancing "capsule" • Nonperipheral "washout" • Threshold growth | One | LR-3 | LR-4 | LR-4 | LR-4 / LR-5 | LR-5 |
| | ≥ Two | LR-4 | LR-4 | LR-4 | LR-5 | LR-5 |

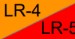 Observations in this cell are categorized based on one additional major feature:
• LR-4 – if enhancing "capsule"
• LR-5 – if nonperipheral "washout" **OR** threshold growth

Figure 3: Li-RADS v2018 (Chernyak et al., 2018)

Motivated by these rules, we define a differentiable "soft" LR-5 score $p_i^{\mathrm{LR\text{-}5,rule}}$ that combines the predicted probabilities of APHE, washout, capsule and TG with the lesion diameter through smooth probabilistic logical operators and diameter-dependent gates. The resulting soft LR-5 probability closely mimics the original Li-RADS decision table while remaining fully differentiable, and is used as an additional supervision signal for the LR-5 head during training. A complete mathematical specification of the probabilistic operators, diameter gates and branch combinations is provided in Appendix E.

### 2.5.4. Multi-task loss with uncertainty-based weighting

We jointly optimise the four binary concept heads (APHE, washout, capsule, LR-5) with a binary cross-entropy (BCE) loss and the seven soft-label components with a mean squared error (MSE) loss. For a logit $z_i^k$ and target $y_i^k \in [0, 1]$ of concept $k \in \{a, w, c, \ell\}$, and for soft-label component $j \in \{1, \ldots, 7\}$ with ground truth $s_i^j$ and prediction $\hat{s}_i^j$, the per-task losses are

$$L_k = \frac{1}{N} \sum_{i=1}^{N} \ell_{\mathrm{BCE}}(z_i^k, y_i^k), \qquad L_j^{\mathrm{soft}} = \frac{1}{N} \sum_{i=1}^{N} (\hat{s}_i^j - s_i^j)^2$$

To automatically balance these heterogeneous tasks, we adopt the uncertainty-based weighting scheme of Kendall et al. (Cipolla et al., 2018). Each task $t$ (concept or soft-label component) has a learned uncertainty parameter $\tau_t$, leading to the global multi-task loss

$$\mathcal{L} = \sum_t \left( e^{-2\tau_t} L_t + \tau_t \right).$$

We also add a gradient-based border regularisation term that penalises sensitivity to pixels at the crop borders. Let $\partial\Omega$ denote the set of border pixels of the input crop $x$; the border penalty and final objective are

$$\mathcal{L}_{\mathrm{border}} = \frac{1}{|\partial\Omega|} \sum_{p \in \partial\Omega} \left\| \nabla_{x_p} \mathcal{L} \right\|_2^2, \qquad \mathcal{L}_{\mathrm{total}} = \mathcal{L} + \lambda_{\mathrm{border}} \mathcal{L}_{\mathrm{border}}$$

Where $\lambda_{\mathrm{border}}$ was set to 0.1. More details are available in Appendix F.

## 2.6. Network Architecture

### 2.6.1. Single-head baseline CNN

As a baseline, we use a single-head CNN that directly predicts HCC from the image. It is based on EfficientNet-B0 pre-trained on ImageNet. The first convolutional layer is adapted from 3 to $C_{\mathrm{in}} = 25$ input channels (5 slices from native, arterial, venous and delayed phase, and lesion mask) to accommodate our multi-phase 2.5D MRI input by copying the pre-trained RGB filters and initialising the additional channels with their mean. The original classifier is replaced by a single fully connected layer with one output logit for LR-5, applied to the global EfficientNet embedding. This model denoted *I (1 head)* therefore ignores Li-RADS concepts and soft labels and is trained only with the single-task BCE loss.

### 2.6.2. Multiple-heads Li-RADS concept bottleneck network

For our Li-RADS concept bottleneck models, we replace the final classifier of the single-head baseline with an identity layer so that the backbone outputs a global feature vector $\mathbf{f}_i \in \mathbb{R}^D$ for each lesion. On top of this shared representation, a regression head (MLP) predicts seven continuous soft-label scores $\hat{\mathbf{s}}_i \in \mathbb{R}^7$ encoding phase-wise and inter-phase intensity deltas for lesion–parenchyma and capsule contrast (Section 2.5).

The concept heads then predict APHE, washout and capsule in a hierarchical manner based on the soft-label predictions: APHE is predicted from the arterial lesion–parenchyma delta, washout from all parenchymal and washout deltas together with the APHE logit, and

capsule from the rim-related deltas and the APHE logit, yielding the concept logit vector $\mathbf{z}_i = (z_i^a, z_i^w, z_i^c)$. A soft LR-5 probability $p_i^\ell$ is finally obtained by combining the predicted concept probabilities with lesion morphology (diameter and threshold growth) through the differentiable Li-RADS rule of Section 2.5.3. More details are provided in Appendix G.

## 2.7. Experiments

We compare three configurations that differ in the number of prediction heads and in the use of lesion morphology (LM) and Li-RADS–inspired soft labels (SL), starting from the imaging input (I): *I (1 head)*, *I + LM (3 heads)*, and *I + LM + SL (4 heads)* .

**Baseline: *I (1 head)*.** In the *I (1 head)* baseline, the classifier operates solely on the imaging embedding extracted from the multi-phase 2.5D MRI input to predict the LR-5 labels using a binary cross-entropy loss.

**Concept bottleneck variants.** The multi-head configurations build on the Li-RADS concept bottleneck architecture described in Section 2.6.2. Starting from the shared EfficientNet backbone, we add:

1. a regression head that predicts the Li-RADS–inspired soft labels,

2. three classification heads that predict the APHE, washout, and capsule concepts, used for LR-5 prediction through the differentiable Li-RADS rule.

These heads are trained in a multi-task setting using the uncertainty-weighted loss introduced in Section 2.5.4, combining losses for APHE, washout, capsule, LR-5 classification, and soft-label regression.

**Main experiment: *I + LM + SL (4 heads)*.** In the main experiment, we evaluate the full *I + LM + SL (4 heads)* configuration. In addition to the imaging embedding, the network receives lesion morphology features (diameter and threshold growth), and it is supervised with Li-RADS–inspired soft labels. The contrast/regression and concept heads are trained to predict the phase- and feature-specific soft labels, while the LR-5 head is trained on the binary HCC label. All heads share the same backbone and are optimized jointly.

**Ablation: *I + LM (3 heads)*.** To isolate the effect of the soft-label supervision, we also evaluate an *I + LM (3 heads)* configuration. In this ablation, we retain the imaging and lesion morphology inputs but remove the optimisation of the regression head with soft labels, so that only the APHE/washout/capsule concept heads and the LR-5 head contribute to the training objective.

### 2.7.1. TRAINING AND OPTIMIZATION

All models were trained with the Adam optimizer (learning rate $1 \times 10^{-4}$, weight decay $10^{-5}$) for up to 100 epochs. We used early stopping based on the validation performance, stopping training if the validation metric did not improve for 15 consecutive epochs.

Model selection was based on the best average validation ROC AUC across tasks (LR-5 prediction and the Li-RADS concept outputs for multi-task models), and all test-set metrics reported in the paper were computed using this selected checkpoint.

### 2.7.2. Metrics

All evaluations are performed at the lesion level. For LR-5 prediction (HCC vs non-HCC) and Li-RADS concepts (APHE, washout, capsule), we report ROC AUC, PR AUC and Matthews correlation coefficient (MCC) to account for class imbalance. In cross-validation, metrics are computed on each validation fold and summarised as mean $\pm$ standard deviation; on the held-out test set, lesion-wise metrics are computed using the selected checkpoint. The same metrics are used in the concept-intervention experiments, where we evaluate the model under simulated edits of the Li-RADS concepts following the intervention principle introduced by (Koh et al., 2020).

To assess explanation quality, we report (i) explanation accuracy, measuring the proportion of NormGrad (Rebuffi et al., 2019) saliency within the lesion mask, and (ii) explanation stability, quantifying the consistency of saliency maps and predictions under small geometric and intensity perturbations, following stability-based XAI evaluation (Raatikainen and Rahtu, 2025). More details are available in Appendix H.

Statistical significance was assessed using paired tests on pooled out-of-fold results from 3-fold cross-validation (each sample evaluated once on its held-out fold). For continuous explainability metrics, we used a two-sided Wilcoxon signed-rank test on paired per-sample values. For binary outcomes (e.g., Pointing Game hit/miss and thresholded LR-5 decisions), we used McNemar's test on paired outcomes (discordant pairs). All tests are two-sided with $\alpha = 0.05$.

## 3. Results

**Trade-off between explanation robustness and LR-5 performance.** Table 1 shows that adding Li-RADS concepts and lesion morphology substantially strengthens explanations. Compared to the single-head model (I, 1 head), the I + LM (3 heads) architecture achieves higher explanation accuracy and geometric stability, and reduces intensity instability by about one order of magnitude, for only a modest decrease in PR AUC and MCC. The 4-head variant with soft contrast surrogates (I + LM + SL) significantly lowers LR-5 metrics compared to I + LM, but provides the most contrast-robust behavior.

Table 1: LR-5 classification performance (definite HCC vs non-HCC).

| | | Explainability | | | | Classification | |
| --- | --- | --- | --- | --- | --- | --- | --- |
| | Expl. Acc.(%)↑ | Geo. Stab.↑ | Int. Stab.↓ | Pointing Game↑ | ROC AUC↑ | PR AUC↑ | MCC↑ |
| **Validation** (3-fold CV; $N_l$=67, 69, 59; $HCC$=44, 55, 44) | | | | | | | |
| I (1 head) | 67.45±13.45 | 0.61±0.09 | 0.12±0.02 | 69.42 ± 11.12 | **0.80±0.05** | **0.89±0.08** | **0.52±0.06** |
| I + LM (3 heads) | **73.07±16.09** | **0.71±0.03** | 0.03±0.01 | **76.38 ± 16.81** | 0.71±0.02 | 0.88±0.03 | 0.38±0.03 |
| I + LM + SL (4 heads) | 56.56±15.19 | 0.65±0.11 | **0.01±0.01** | 59.09 ± 17.12 | 0.74±0.01 | 0.88±0.03 | 0.42±0.05 |
| **Test** ($N_l$=55; $HCC$=38) | | | | | | | |
| I (1 head) | 59.67±9.79 | 0.55±0.08 | 0.13±0.02 | 64.85 ± 7.62 | **0.81±0.03** | 0.90±0.03 | **0.55±0.03** |
| I + LM (3 heads) | **73.07±11.16**** | **0.69±0.06*** | 0.03±0.01** | **81.21 ± 10.43** | 0.80±0.05 | **0.91±0.02** | 0.53±0.08 |
| I + LM + SL (4 heads) | 62.16±15.17 | 0.65±0.10 | **0.01±0.01**** | 69.70 ± 16.42 | 0.74±0.03 | 0.88±0.02 | 0.42±0.04† |

Best per column in **bold**. * $p < 0.05$, ** $p < 0.01$ (two-sided) better than baseline; † $p < 0.05$ worse than baseline. *Abbreviations:* AUC = area under the ROC curve; SL = Soft labels; LM = Lesion morphology; $N_l$ = number of lesions; HCC = hepatocellular carcinoma.

**Effect of physics-driven soft labels on Li-RADS concepts.** Table 2 compares two ways of handling Li-RADS concepts. Both models predict seven continuous contrast surrogates, but only the 4-head variant receives explicit supervision on these surrogates and then derives APHE, washout and capsule from them, whereas the 3-head model is directly supervised on the three concepts only. On validation, this surrogate factorisation benefits APHE (higher ROC/PR AUC and MCC). However, on the held-out test set it underperforms for washout and capsule, while APHE cannot be reported because the test set contains only APHE-positive lesions. On the external cohort ($N_l$=10), performance estimates are imprecise, and we do not observe a consistent transfer of the APHE benefit; direct concept supervision remains higher across all three concepts. Overall, these results suggest that simple global intensity deltas can provide a useful in-domain inductive bias for APHE, but washout and capsule likely rely on more complex temporal and boundary cues than captured by our current surrogates.

**Contrast-surrogate supervision ablation.** Table 3 (Appendix I) disentangles which surrogate components drive Li-RADS concept prediction. Across internal validation, the main benefit of SL is concentrated on APHE, and is largely driven by parenchymal contrast deltas, while adding additional washout/ring surrogates yields diminishing or inconsistent gains. In contrast, washout and capsule do not consistently benefit from SL: the strongest in-domain performance is often achieved without SL, and on the held-out test set ring-based surrogates can improve washout discrimination without translating into a consistent MCC gain. On the external cohort, trends are noisier due to the small sample size and the APHE advantage does not consistently transfer; washout/capsule remain sensitive to the chosen surrogate subset. Overall, SL acts as an optional, concept- and domain-dependent regularizer—useful for APHE in-domain, but requiring careful selection (or omission) for washout and capsule.

However, these findings are consistent with reported inter-reader variability for Li-RADS major features (Hong et al., 2023), with intraclass correlation coefficients of 0.65, 0.50 and 0.50 for APHE, washout and capsule, respectively. APHE labels are more reliable, which aligns with the improved APHE performance observed when using contrast surrogates. Conversely, washout and capsule are more difficult to assess and exhibit higher annotation variability, which may also explain why introducing soft surrogates sometimes leads to worse predictions for these concepts, reflecting an underlying uncertainty in radiologist labelling.

**Li-RADS concept interventions and human-in-the-loop use.** We additionally evaluated the 4-head model under simulated concept-level interventions, progressively replacing predicted APHE, washout and capsule labels with their ground truth before recomputing the LR-5 decision. On the test set, ROC AUC improved from 0.74 without intervention to 0.96 with 50–100 corrected concepts, while MCC increased from 0.42 to 0.91 and F1 from 0.84 to 0.97, indicating that a relatively small number of concept edits can substantially boost LR-5 classification without retraining. Full results and experimental details are provided in Appendix J (Table S4).

Figure 4 illustrates a prediction from the 4-head model in a format that could be presented to a radiologist. The top row displays the lesion in the four dynamic phases with the lesion contour; the bottom row shows gradient-based attribution maps for each Li-RADS concept (APHE, washout, capsule). On the right, a diagnostic panel summarizes clinical

Table 2: Li-RADS concept prediction performance.

| | | ROC AUC↑ | PR AUC↑ | MCC↑ |
|---|---|---|---|---|
| **Validation** | | | | |
| (3-folds CV; $N_l$=67, 69, 59; $HCC = 44, 55, 44$) | | | | |
| APHE | I + LM (3 heads) | 0.73±0.05 | 0.98±0.01 | 0.28±0.03 |
| | I + LM + SL (4 heads) | **0.81±0.07** | **0.99±0.01** | **0.40±0.09** |
| Washout | I + LM (3 heads) | **0.73±0.05** | 0.87±0.05 | **0.45±0.06** |
| | I + LM + SL (4 heads) | 0.68±0.05 | **0.83±0.06** | 0.40±0.07 |
| Capsule | I + LM (3 heads) | **0.80±0.04** | **0.79±0.11** | **0.56±0.05** |
| | I + LM + SL (4 heads) | 0.61±0.03 | 0.69±0.04 | 0.32±0.01 |
| **Test** $(N_l = 55; HCC = 38)$ | | | | |
| APHE | I + LM (3 heads) | - | - | - |
| | I + LM + SL (4 heads) | - | - | - |
| Washout | I + LM (3 heads) | **0.69±0.03** | **0.83±0.01** | **0.45±0.06** |
| | I + LM + SL (4 heads) | 0.58±0.07 | 0.79±0.04 | 0.24±0.06 |
| Capsule | I + LM (3 heads) | **0.87±0.03** | **0.82±0.05** | **0.65±0.07** |
| | I + LM + SL (4 heads) | 0.66±0.08 | 0.63±0.08 | 0.41±0.10 |
| **External test** $(N_l = 10; HCC = 5)$ | | | | |
| APHE | I + LM (3 heads) | **0.58±0.41** | **0.85±0.18** | **0.51±0.36** |
| | I + LM + SL (4 heads) | 0.50±0.27 | 0.82±0.14 | 0.34±0.26 |
| Washout | I + LM (3 heads) | **0.41±0.27** | **0.56±0.18** | **0.30±0.22** |
| | I + LM + SL (4 heads) | 0.28±0.09 | 0.46±0.09 | 0.22±0.16 |
| Capsule | I + LM (3 heads) | **0.65±0.13** | **0.67±0.15** | **0.54±0.10** |
| | I + LM + SL (4 heads) | 0.55±0.19 | 0.62±0.18 | 0.44±0.08 |

Best per concept in **bold**. *Abbreviations:* AUC = area under the ROC curve; SL = Soft labels, $N_l$ = Number of lesions, HCC = Hepatocellular Carcinoma. APHE performance cannot be reported on the test set because it contains only APHE-positive lesions.

features (diameter, growth), concept and LR-5 probabilities, and the quantitative contrast surrogates. This layout allows the radiologist to check whether explanations focus on the lesion, to relate each concept probability to its soft contrast measurements, and to understand how the model arrived at its LR-5 score. Comparisons with the other models and example of predictions are provided in Appendix K including representative success and failure cases, illustrating when explanations are misleading despite confident predictions, motivating cautious use as decision support and the need for prospective human-centered evaluation.

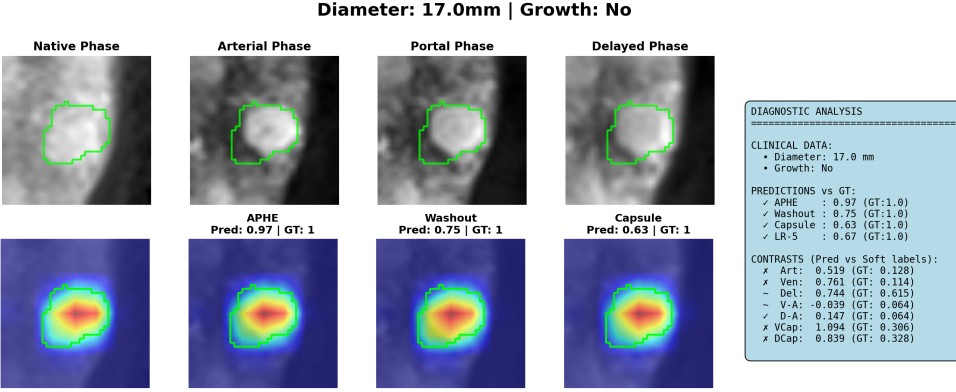

Figure 4: Example of prediction with the I + LM + SL (4 heads) model

## 4. Conclusion

We introduced a clinically grounded, Li-RADS–aligned diagnostic support system for HCC characterization on DCE-MRI, which produces the final LR-5 decision together with clinically relevant intermediate cues (e.g., APHE, washout, capsule). This design aims to make model outputs more transparent and actionable by exposing clinically meaningful evidence that can be inspected and, when needed, corrected.

Importantly, we emphasize that improved explanation metrics do not guarantee improved diagnostic performance. In practice, explanation-oriented constraints can introduce trade-offs: concept bottlenecks may discard task-relevant information not captured by the selected concepts, and multi-task concept supervision can lead to negative transfer when some concepts are noisily supervised. Consistent with this, our contrast-surrogate soft-label ablation (Table 3) shows that soft labels are beneficial for APHE but can be neutral or detrimental for washout/capsule, motivating concept-dependent use of such supervision. Moreover, stabilizing explanations (e.g., saliency maps) may come at the cost of reduced fidelity to the underlying decision signal, and human factors such as automation bias can further affect real-world performance if users over-rely on AI outputs. A further limitation is that our experiments rely on expert lesion masks (used to define ROIs and provided as an input channel); in deployment, these masks would need to be generated by automated segmentation/detection models, and quantifying robustness to such masks is an important next step. Therefore, a key next step is a dedicated reader study comparing baseline outputs vs. our concept-based interface, measuring diagnostic accuracy, time-to-decision, and inter-/intra-reader reproducibility, alongside continued refinement of concept supervision for the more challenging Li-RADS features.

## Acknowledgments

This work was supported by the grant 320030_207944 provided by the Swiss National Science Foundation (SNF).

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

## Appendix A. Image Parameters and Analysis

All MR examinations followed standard clinical liver MRI protocols aligned with Li-RADS technical recommendations. When histopathology was available, it served as the reference standard; otherwise, diagnoses were based on imaging criteria per Li-RADS v2018. For this study, we analyzed axial T1-weighted images (pre-contrast) and the dynamic post-contrast series: late arterial, portal venous, and delayed phases acquired after intravenous administration of an extracellular gadolinium agent (gadoterate meglumine / gadoteric acid) at 20 sec, 3 min and 5 min.

Volumetric liver masks were generated automatically using a pre-trained nnU-Net model, while lesion masks were manually delineated and reviewed by an abdominal radiologist (Monnin et al., 2025). Lesion segmentations were performed independently across all relevant sequences by a radiology resident using full patient context (including pathology and follow-up imaging when available) as reference. All segmentations were created in dedicated clinical software (Mint Lesion$^{TM}$) and subsequently reviewed for quality and consistency by an abdominal radiologist with eight years of experience, who also assigned Li-RADS categories to each lesion. While all HCC-positive patients had at least one confirmed HCC, additional non-HCC hepatic lesions were included for analysis.

## Appendix B. Background suppression transform

To further reduce shortcut learning on non-hepatic background regions, we implemented a custom background suppression transform applied during training. The transform operates on the image, lesion mask and liver mask and is applied with probability $p_{\mathrm{bg}} = 0.8$.

Let $I \in \mathbb{R}^{C \times H \times W \times D}$ denote the multi-phase lesion crop, $M_{\mathrm{liver}}$ the binary liver mask and $M_{\mathrm{lesion}}$ the binary lesion mask (both broadcast to the same shape as $I$). We first construct a combined foreground mask

$$M_{\mathrm{fg}} = \mathrm{clip}\big(M_{\mathrm{liver}} + M_{\mathrm{lesion}},\, 0,\, 1\big),$$

and its complement $M_{\mathrm{bg}} = 1 - M_{\mathrm{fg}}$, which identifies non-hepatic background voxels.

At each training iteration, with probability $p_{\mathrm{bg}}$ we draw a background suppression mode $m \in \{\mathrm{mask}, \mathrm{smooth}, \mathrm{noise}\}$ according to

$$\mathbb{P}(m = \mathrm{mask}) = 0.4, \quad \mathbb{P}(m = \mathrm{smooth}) = 0.4, \quad \mathbb{P}(m = \mathrm{noise}) = 0.2.$$

The transform is then applied channel-wise as follows:

- **Masking** ($m = \mathrm{mask}$): background voxels are fully suppressed by setting them to zero,
$$I' = I \odot M_{\mathrm{fg}}.$$

- **Smooth suppression** ($m = \mathrm{smooth}$): background intensities are attenuated by a random factor $\alpha \sim \mathcal{U}(0.1, 0.3)$,
$$I' = I \odot \big(M_{\mathrm{fg}} + (1 - \alpha)\, M_{\mathrm{bg}}\big).$$

- **Noise** ($m = \mathrm{noise}$): in our experiments, we did not systematically exploit this mode and therefore only considered the masking and smooth suppression cases reported above.

If the transform is not applied (with probability $1 - p_{\mathrm{bg}}$), we simply set $I' = I$. The modified image $I'$ is then passed through the remaining MONAI transforms and the network.

Intuitively, this background suppression encourages the model to focus on intra-hepatic lesion and liver appearance, while reducing the incentive to rely on spurious patterns in the surrounding background (e.g., edges of the crop or extra-hepatic anatomy).

## Appendix C. Image pre-processing and 2.5D lesion crops

All multiphase liver MRI volumes were first normalised by dividing each phase by the median signal of the erector spinae muscles for that patient. For model training, we then applied a MONAI-based 3D preprocessing pipeline to the lesion-centred image, lesion mask and liver mask: random in-plane flips (probability 0.5 per axis), random 90° rotations (probability 0.5), a random affine transform (probability 0.5; rotation range ±30°, isotropic scaling in $[0.85, 1.15]$), and random zoom (probability 0.5; zoom factor in $[0.8, 1.2]$). Volumes were resampled to a fixed in-plane size of $224 \times 224$ voxels while preserving the through-plane dimension. To reduce shortcut learning on extra-hepatic background, we applied a background suppression transform to the image with probability 0.8, randomly masking or attenuating voxels outside the union of liver and lesion masks (see Appendix B), followed by random Gaussian noise and smoothing. Finally, image intensities were normalised on non-zero voxels.

For each annotated lesion, we extracted a 3D crop around the lesion and identified all axial slices with non-zero segmentation. If the lesion covered at least five slices, we selected five consecutive slices centred on the middle lesion slice; otherwise, the first and last lesion slices were replicated symmetrically to obtain exactly five slices. The same indices were used to extract the corresponding liver and lesion masks, yielding a 2.5D multi-phase crop with four contrast phases (native, arterial, venous, delayed) over five slices, which were concatenated with the lesion mask and served as input to the classifier.

In addition, for the computation of Li-RADS–inspired soft labels only, we constructed auxiliary phase-difference images by subtracting the native phase from each dynamic phase (arterial, venous, delayed) and applying per-lesion min–max normalisation on voxels $> 0$, with values clipped to $[0, 1]$. These difference images were not used as input to the network and were only fed to the soft-label computation.

## Appendix D. Details on Li-RADS–inspired soft labels

This appendix visually illustrates and complements the computation of the Li-RADS–inspired soft labels described in Section 2.5. Soft labels are computed on 3D lesion-centred patches with four channels (native, arterial, venous, delayed). For each connected lesion component $K_i$ inside the liver mask $L$, we derive lesion, peri-lesional ring and parenchyma regions $K_i, R_i, P_i$ as in Section 2.5, and work on phase-difference images $I_p$ for $p \in \{1, 2, 3\}$ (arterial, venous, delayed).

**Quantile-based robust medians.** Inside the lesion, we compute phase-specific medians $m_{p,i}^{\text{in}}$ after discarding extreme values by restricting to fixed quantile ranges: for the arterial phase, we keep intensities between the 50th and 100th percentiles; for the venous and delayed phases, we keep intensities between the 0th and 50th percentiles. In the peri-lesional ring $R_i$, venous and delayed intensities are summarised by medians computed between approximately the 50th and 100th percentiles, emphasising brighter rim voxels that are most relevant for capsule-like enhancement. In the parenchyma region $P_i$, medians $m_{p,i}^{\text{par}}$ are computed from central liver intensities using a fixed interquartile range (between the 25th and 75th percentiles) to obtain robust estimates of background liver signal.

From these medians, we form the phase-wise lesion–parenchyma deltas

$$d_{p,i}^{\text{par}} = m_{p,i}^{\text{in}} - m_{p,i}^{\text{par}}, \quad p = 1, 2, 3,$$

washout-related deltas

$$d_{p,i}^{\text{wash}} = d_{p,i}^{\text{par}} - d_{1,i}^{\text{par}}, \quad p = 2, 3,$$

and rim deltas

$$d_{p,i}^{\text{ring}} = m_{p,i}^{\text{in}} - m_{p,i}^{\text{ring}}, \quad p = 2, 3,$$

which are assembled into the 7D soft-label vector

$$\mathbf{s}_i = \left( d_{1,i}^{\text{par}}, d_{2,i}^{\text{par}}, d_{3,i}^{\text{par}}, d_{2,i}^{\text{wash}}, d_{3,i}^{\text{wash}}, d_{2,i}^{\text{ring}}, d_{3,i}^{\text{ring}} \right),$$

used as regression target for the contrast head. An example of how lesion, parenchyma and peri-lesional ring voxels are selected to compute these medians and deltas is illustrated in Fig. 5.

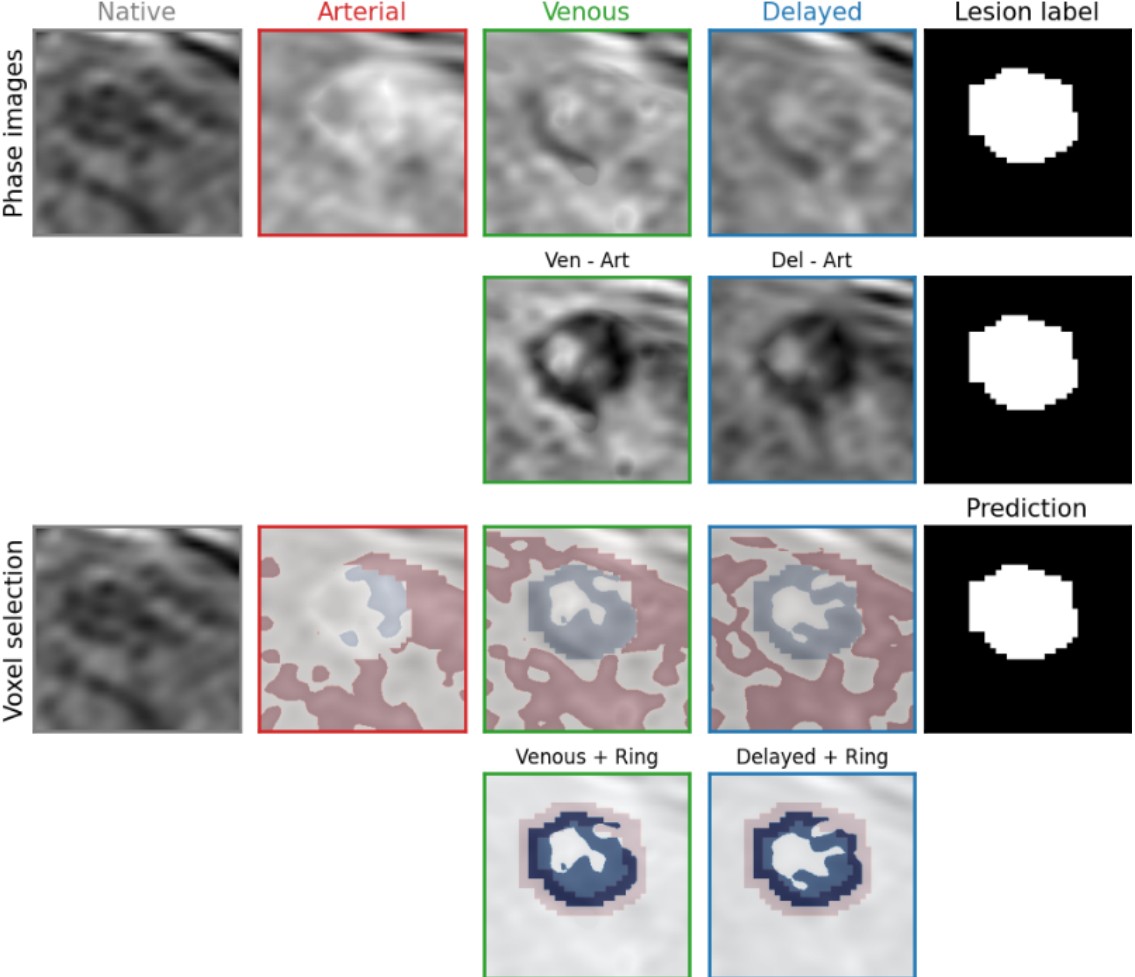

Figure 5: Illustration of the computation of Li-RADS–inspired soft labels for one example lesion. **Top row (Phase images):** native, three dynamic phases (1–3) and the binary lesion mask. **Second row (Phase differences):** phase-difference images highlighting temporal washout (phase 2 minus phase 1, phase 3 minus phase 1). **Third row (Voxel selection):** example of voxel selection used to compute lesion–parenchyma contrasts, with lesion voxels and surrounding liver parenchyma highlighted. **Bottom row (Ring selection):** venous and delayed phases with a peri-lesional ring used to derive capsule-related rim contrast. These regions provide the statistics that define the seven continuous soft labels $\mathbf{s}_i$.

## Appendix E. Differentiable Implementation of Li-RADS

The LR-5 prediction is obtained with a soft, differentiable approximation of the Li-RADS LR-5 rule. We use probabilistic logical operators

$$\mathrm{OR}_p(x_1, \ldots, x_n) = 1 - \prod_{j=1}^{n}(1 - x_j), \qquad \mathrm{AND}_p(a, b) = a\,b,$$

and diameter-dependent gates based on the lesion diameter $d_i$ (in mm):

$$g_{10\text{–}19}(d_i) = \sigma\big(k(d_i - 10)\big)\Big(1 - \sigma\big(k(d_i - 19)\big)\Big), \qquad g_{\geq 20}(d_i) = \sigma\big(k(d_i - 20)\big),$$

with $k > 0$ a steepness parameter (here $k = 3$).

Let $p_i^{\mathrm{tg}} \in [0, 1]$ denote the (soft) indicator of threshold growth. The 10–19 mm branch encodes APHE $\wedge$ (washout $\vee$ TG):

$$r_{10\text{–}19,i} = g_{10\text{–}19}(d_i)\,\mathrm{AND}_p\Big(p_i^a,\,\mathrm{OR}_p(p_i^w, p_i^{\mathrm{tg}})\Big),$$

while the $\geq 20$ mm branch encodes APHE $\wedge$ (washout $\vee$ capsule $\vee$ TG):

$$r_{\geq 20,i} = g_{\geq 20}(d_i)\,\mathrm{AND}_p\Big(p_i^a,\,\mathrm{OR}_p(p_i^w, p_i^c, p_i^{\mathrm{tg}})\Big).$$

The final soft LR-5 probability is

$$p_i^{\mathrm{LR\text{-}5,rule}} = \mathrm{OR}_p\big(r_{10\text{–}19,i}, r_{\geq 20,i}\big) = 1 - \big(1 - r_{10\text{–}19,i}\big)\big(1 - r_{\geq 20,i}\big),$$

which is used as a differentiable supervision signal for the LR-5 head.

## Appendix F. Multi-task loss with uncertainty-based weighting

Given the multi-head architecture described, we optimize the concept classification and soft label regression tasks jointly. For the four binary concept outputs (APHE, washout, capsule, LR-5), we use a binary cross-entropy (BCE) loss with logits. For a single logit $z$ and label $y \in [0,1]$, the average BCE losses over the batch for each concept and for LR-5 are

$$L_k = \frac{1}{N} \sum_{i=1}^{N} \ell_{\mathrm{BCE}}(z_i^k, y_i^k), \qquad k \in \{a, w, c, \ell\},$$

For each soft-label component of the regression head, we define a separate MSE loss, with $\mathbf{s}_i = (s_i^1, \ldots, s_i^7)$ and $\hat{\mathbf{s}}_i = (\hat{s}_i^1, \ldots, \hat{s}_i^7)$ denoting the ground-truth and predicted soft-label vectors, respectively. Then, for each component $j \in \{1, \ldots, 7\}$,

$$L_j^{\mathrm{soft}} = \frac{1}{N} \sum_{i=1}^{N} (\hat{s}_i^j - s_i^j)^2,$$

with the seven indices $j$ corresponding to components of $\hat{s}$ (see section 2.5.2).

To automatically balance the contribution of the different tasks, we use the uncertainty-based weighting scheme of Kendall et al. (Cipolla et al., 2018). This approach interprets each task-specific weight as the inverse of a (homoscedastic) uncertainty term and has two main advantages in practice: (i) it avoids manual tuning of heuristic loss weights, and (ii) it down-weights tasks that are intrinsically noisier or harder to fit, stabilising multi-task training.

We introduce one uncertainty parameter for each classification task (APHE, washout, capsule, LR-5), and a vector of uncertainty parameters for the seven soft-label regression components,

$$\theta_a, \ \theta_w, \ \theta_c, \ \theta_\ell, \qquad \boldsymbol{\tau} = \left(\tau_1^{\mathrm{par}}, \tau_2^{\mathrm{par}}, \tau_3^{\mathrm{par}}, \tau_2^{\mathrm{wash}}, \tau_3^{\mathrm{wash}}, \tau_2^{\mathrm{ring}}, \tau_3^{\mathrm{ring}}\right),$$

so that each phase- and feature-specific delta has its own learned uncertainty.

Let $\mathcal{T}_{\mathrm{cls}} = \{a, w, c, \ell\}$ denote the classification tasks, with the uncertainty parameters $\theta_a, \theta_w, \theta_c, \theta_\ell$, and let $\mathcal{T}_{\mathrm{reg}} = \{(1, \mathrm{par}), (2, \mathrm{par}), (3, \mathrm{par}), (2, \mathrm{wash}), (3, \mathrm{wash}), (2, \mathrm{ring}), (3, \mathrm{ring})\}$ denote the soft-label regression components, with uncertainty parameters $\tau_1^{\mathrm{par}}, \ldots, \tau_3^{\mathrm{ring}}$. For each task $t \in \mathcal{T} = \mathcal{T}_{\mathrm{cls}} \cup \mathcal{T}_{\mathrm{reg}}$, we denote its loss by $L_t$ and its uncertainty parameter by $\tau_t$ and describe the total loss as

$$\mathcal{L} = \sum_{t \in \mathcal{T}} \left( \exp(-2\tau_t)\, L_t + \tau_t \right).$$

We also add a gradient-based border regularization term to discourage spurious reliance on image borders. Let $\mathcal{L}$ denote the uncertainty-weighted multi-task loss defined above and $\mathcal{L}_{\mathrm{border}}$ the corresponding border-gradient penalty. The final training objective is

$$\mathcal{L}_{\mathrm{total}} = \mathcal{L} + \lambda_{\mathrm{border}}\, \mathcal{L}_{\mathrm{border}}$$

Concretely, $\mathcal{L}_{\text{border}}$ penalizes the squared norm of the input gradient at the crop borders,

$$\mathcal{L}_{\text{border}} = \frac{1}{|\partial\Omega|} \sum_{p \in \partial\Omega} \left\| \nabla_{x_p} \mathcal{L} \right\|_2^2,$$

where $\partial\Omega$ denotes the set of border pixels of the input crop $x$ and $\lambda_{\text{border}} > 0$ controls the strength of the regularization. For all experiments, $\lambda_{\text{border}}$ was set to 0.1.

## Appendix G. Li-RADS Concept Bottleneck

For our Li-RADS concept bottleneck models, we modify the single-head baseline as follows: the original classifier layer is replaced by an identity mapping, so that the backbone outputs a global feature vector $\mathbf{f}_i \in \mathbb{R}^D$ for each lesion $i$. A first regression head then predicts a 7D soft-label vector $\hat{\mathbf{s}}_i \in \mathbb{R}^7$ from $\mathbf{f}_i$ via a small MLP (Linear, GELU, LayerNorm, Dropout, Linear). These seven dimensions encode the phase-wise lesion–parenchyma deltas and inter-phase differences,

$$\hat{\mathbf{s}}_i = \big(\hat{d}_{1,i}^{\mathrm{par}}, \hat{d}_{2,i}^{\mathrm{par}}, \hat{d}_{3,i}^{\mathrm{par}}, \hat{d}_{2,i}^{\mathrm{wash}}, \hat{d}_{3,i}^{\mathrm{wash}}, \hat{d}_{2,i}^{\mathrm{ring}}, \hat{d}_{3,i}^{\mathrm{ring}}\big),$$

as described in Section 2.5.

On top of these soft labels, three concept heads predict the Li-RADS major features in a hierarchical manner. The APHE head takes as input only the arterial delta $\hat{d}_{1,i}^{\mathrm{par}}$ and outputs an APHE logit $z_i^a$. The washout head then operates on the five deltas $\hat{d}_{1,i}^{\mathrm{par}}, \hat{d}_{2,i}^{\mathrm{par}}, \hat{d}_{3,i}^{\mathrm{par}}, \hat{d}_{2,i}^{\mathrm{wash}}, \hat{d}_{3,i}^{\mathrm{wash}}$ concatenated with the APHE logit $z_i^a$ to produce a washout logit $z_i^w$. Finally, the capsule head takes as input the two rim-related deltas $\hat{d}_{2,i}^{\mathrm{ring}}, \hat{d}_{3,i}^{\mathrm{ring}}$ and the APHE logit $z_i^a$ to produce a capsule logit $z_i^c$. Collecting these three logits yields the concept vector

$$\mathbf{z}_i = \big(z_i^a, z_i^w, z_i^c\big),$$

corresponding to APHE, washout and capsule, respectively, with probabilities $p_i^k = \sigma(z_i^k)$ for $k \in \{a, w, c\}$.

The LR-5 score is not predicted by an additional free classifier. Instead, we compute a soft LR-5 probability $p_i^\ell$ by combining the concept probabilities $(p_i^a, p_i^w, p_i^c)$ with lesion morphology measurements (diameter and threshold growth) through a differentiable approximation of the Li-RADS decision rule, as described in Section 2.5.3.

## Appendix H. Metrics

All evaluations are performed at the lesion level. For LR-5 prediction (HCC vs non-HCC) and for the Li-RADS concept outputs (APHE, washout, capsule), we quantify discrimination using the area under the receiver-operating characteristic curve (ROC AUC) and the area under the precision–recall curve (PR AUC), treating presence of the feature or HCC as the positive class. Given the class imbalance in our cohorts, we additionally report the Matthews correlation coefficient (MCC), which summarizes the full confusion matrix and is more informative than accuracy or $F_1$ in imbalanced binary classification.

For the cross-validation experiments, metrics are computed separately on each validation fold and summarised as mean $\pm$ standard deviation over the three folds. On the held-out test set, we report lesion-wise metrics computed on all test lesions, using the checkpoint selected as described in Section 2.7.1. The same set of metrics (ROC AUC, PR AUC, MCC) is used in the concept-intervention experiments, where we evaluate LR-5 performance as a function of the proportion of corrected concept predictions.

To assess the quality of the explanations provided by NormGrad, we report two complementary metrics: explanation accuracy and explanation stability, in line with recent work on stability-based XAI evaluation (Raatikainen and Rahtu, 2025). Explanation accuracy measures how well the saliency maps localize the annotated lesion. For each lesion and each Li-RADS concept, we compute a NormGrad heatmap on the 2.5D input crop, normalise it to sum to one, and record the proportion of total saliency mass falling inside the manual lesion mask. Explanation accuracy is reported as the mean percentage over all lesions and concepts.

Explanation stability measures the robustness of the saliency maps to small perturbations of the input. For each lesion, we generate $K$ NormGrad maps under independent test-time augmentations. A first metric measures the geometric stability by applying 80% center crop with resize and 90° rotation transforms, then computing the mean pairwise Pearson correlation between the flattened, normalized maps. The final stability score is obtained by averaging this correlation over all lesions and concepts, with values closer to 1 indicating more consistent and hence more reliable explanations. We created a second stability metric measuring prediction robustness under intensity variations by applying n=10 random perturbations combining Gaussian noise ($\sigma$=0.05) and intensity scaling uniformly sampled from [0.8,1.2]. We compute the mean absolute difference (MAD) between the original prediction and the mean of perturbed predictions across all tasks. Lower MAD values indicate more robust predictions, with the model maintaining consistent outputs despite realistic imaging variability that can occur during MRI acquisition.

# Appendix I. Soft labels ablation experiment

Table 3: Ablation of contrast-surrogate supervision for LI-RADS concept prediction.

| | ROC AUC↑ | PR AUC↑ | MCC↑ |
|---|---|---|---|
| **Validation** | | | |
| (3-folds CV; $N_l$=67, 69, 59; $HCC = 44, 55, 44$) | | | |
| APHE $\quad d_{1,i}^{\text{par}}, d_{2,i}^{\text{par}}, d_{3,i}^{\text{par}}, d_{2,i}^{\text{wash}}, d_{3,i}^{\text{wash}}, d_{2,i}^{\text{ring}}, d_{3,i}^{\text{ring}}$ | 0.81±0.07 | **0.99±0.01** | 0.40±0.09 |
| $\quad d_{1,i}^{\text{par}}, d_{2,i}^{\text{par}}, d_{3,i}^{\text{par}}, d_{2,i}^{\text{wash}}, d_{3,i}^{\text{wash}}$ | **0.83±0.06** | 0.99±0.01 | **0.48±0.08** |
| $\quad d_{1,i}^{\text{par}}, d_{2,i}^{\text{par}}, d_{3,i}^{\text{par}}, d_{2,i}^{\text{ring}}, d_{3,i}^{\text{ring}}$ | 0.81±0.11 | 0.98±0.02 | 0.46±0.02 |
| $\quad d_{1,i}^{\text{par}}, d_{2,i}^{\text{par}}, d_{3,i}^{\text{par}}$ | 0.77±0.11 | 0.98±0.01 | 0.42±0.20 |
| $\quad$ No SL | 0.73±0.05 | 0.98±0.01 | 0.28±0.03 |
| Washout $\quad d_{1,i}^{\text{par}}, d_{2,i}^{\text{par}}, d_{3,i}^{\text{par}}, d_{2,i}^{\text{wash}}, d_{3,i}^{\text{wash}}, d_{2,i}^{\text{ring}}, d_{3,i}^{\text{ring}}$ | 0.68±0.05 | 0.83±0.06 | 0.40±0.07 |
| $\quad d_{1,i}^{\text{par}}, d_{2,i}^{\text{par}}, d_{3,i}^{\text{par}}, d_{2,i}^{\text{wash}}, d_{3,i}^{\text{wash}}$ | 0.61±0.01 | 0.79±0.05 | 0.26±0.02 |
| $\quad d_{1,i}^{\text{par}}, d_{2,i}^{\text{par}}, d_{3,i}^{\text{par}}, d_{2,i}^{\text{ring}}, d_{3,i}^{\text{ring}}$ | 0.68±0.04 | 0.85±0.02 | 0.32±0.56 |
| $\quad d_{1,i}^{\text{par}}, d_{2,i}^{\text{par}}, d_{3,i}^{\text{par}}$ | 0.71±0.02 | 0.85±0.06 | 0.39±0.05 |
| $\quad$ No SL | **0.73±0.05** | **0.87±0.05** | **0.45±0.06** |
| Capsule $\quad d_{1,i}^{\text{par}}, d_{2,i}^{\text{par}}, d_{3,i}^{\text{par}}, d_{2,i}^{\text{wash}}, d_{3,i}^{\text{wash}}, d_{2,i}^{\text{ring}}, d_{3,i}^{\text{ring}}$ | 0.61±0.03 | 0.69±0.04 | 0.32±0.01 |
| $\quad d_{1,i}^{\text{par}}, d_{2,i}^{\text{par}}, d_{3,i}^{\text{par}}, d_{2,i}^{\text{wash}}, d_{3,i}^{\text{wash}}$ | 0.72±0.10 | 0.77±0.11 | 0.45±0.12 |
| $\quad d_{1,i}^{\text{par}}, d_{2,i}^{\text{par}}, d_{3,i}^{\text{par}}, d_{2,i}^{\text{ring}}, d_{3,i}^{\text{ring}}$ | 0.58±0.08 | 0.53±0.11 | 0.25±0.10 |
| $\quad d_{1,i}^{\text{par}}, d_{2,i}^{\text{par}}, d_{3,i}^{\text{par}}$ | 0.76±0.02 | **0.81±0.06** | 0.51±0.07 |
| $\quad$ No SL | **0.80±0.04** | 0.79±0.11 | **0.56±0.05** |
| **Test** ($N_l = 55$; $HCC = 38$) | | | |
| APHE $\quad d_{1,i}^{\text{par}}, d_{2,i}^{\text{par}}, d_{3,i}^{\text{par}}, d_{2,i}^{\text{wash}}, d_{3,i}^{\text{wash}}, d_{2,i}^{\text{ring}}, d_{3,i}^{\text{ring}}$ | - | - | - |
| $\quad d_{1,i}^{\text{par}}, d_{2,i}^{\text{par}}, d_{3,i}^{\text{par}}, d_{2,i}^{\text{wash}}, d_{3,i}^{\text{wash}}$ | - | - | - |
| $\quad d_{1,i}^{\text{par}}, d_{2,i}^{\text{par}}, d_{3,i}^{\text{par}}, d_{2,i}^{\text{ring}}, d_{3,i}^{\text{ring}}$ | - | - | - |
| $\quad d_{1,i}^{\text{par}}, d_{2,i}^{\text{par}}, d_{3,i}^{\text{par}}$ | - | - | - |
| $\quad$ No SL | - | - | - |
| Washout $\quad d_{1,i}^{\text{par}}, d_{2,i}^{\text{par}}, d_{3,i}^{\text{par}}, d_{2,i}^{\text{wash}}, d_{3,i}^{\text{wash}}, d_{2,i}^{\text{ring}}, d_{3,i}^{\text{ring}}$ | 0.58±0.07 | 0.79±0.04 | 0.24±0.06 |
| $\quad d_{1,i}^{\text{par}}, d_{2,i}^{\text{par}}, d_{3,i}^{\text{par}}, d_{2,i}^{\text{wash}}, d_{3,i}^{\text{wash}}$ | 0.50±0.020 | 0.74±0.01 | 0.18±0.03 |
| $\quad d_{1,i}^{\text{par}}, d_{2,i}^{\text{par}}, d_{3,i}^{\text{par}}, d_{2,i}^{\text{ring}}, d_{3,i}^{\text{ring}}$ | **0.71±0.01** | **0.84±0.04** | 0.41±0.01 |
| $\quad d_{1,i}^{\text{par}}, d_{2,i}^{\text{par}}, d_{3,i}^{\text{par}}$ | 0.68±0.03 | 0.83±0.01 | 0.38±0.07 |
| $\quad$ No SL | 0.69±0.03 | 0.83±0.01 | **0.45±0.06** |
| Capsule $\quad d_{1,i}^{\text{par}}, d_{2,i}^{\text{par}}, d_{3,i}^{\text{par}}, d_{2,i}^{\text{wash}}, d_{3,i}^{\text{wash}}, d_{2,i}^{\text{ring}}, d_{3,i}^{\text{ring}}$ | 0.66±0.08 | 0.63±0.08 | 0.41±0.10 |
| $\quad d_{1,i}^{\text{par}}, d_{2,i}^{\text{par}}, d_{3,i}^{\text{par}}, d_{2,i}^{\text{wash}}, d_{3,i}^{\text{wash}}$ | 0.77±0.04 | 0.73±0.04 | 0.50±0.07 |
| $\quad d_{1,i}^{\text{par}}, d_{2,i}^{\text{par}}, d_{3,i}^{\text{par}}, d_{2,i}^{\text{ring}}, d_{3,i}^{\text{ring}}$ | 0.58±0.09 | 0.53±0.11 | 0.25±0.10 |
| $\quad d_{1,i}^{\text{par}}, d_{2,i}^{\text{par}}, d_{3,i}^{\text{par}}$ | 0.80±0.02 | 0.74±0.03 | 0.55±0.06 |
| $\quad$ No SL | **0.87±0.03** | **0.82±0.05** | **0.65±0.07** |
| **External test** ($N_l = 10$; $HCC = 5$) | | | |
| APHE $\quad d_{1,i}^{\text{par}}, d_{2,i}^{\text{par}}, d_{3,i}^{\text{par}}, d_{2,i}^{\text{wash}}, d_{3,i}^{\text{wash}}, d_{2,i}^{\text{ring}}, d_{3,i}^{\text{ring}}$ | 0.50±0.27 | 0.82±0.14 | 0.34±0.26 |
| $\quad d_{1,i}^{\text{par}}, d_{2,i}^{\text{par}}, d_{3,i}^{\text{par}}, d_{2,i}^{\text{wash}}, d_{3,i}^{\text{wash}}$ | 0.52±0.21 | 0.86±0.07 | 0.44±0.17 |
| $\quad d_{1,i}^{\text{par}}, d_{2,i}^{\text{par}}, d_{3,i}^{\text{par}}, d_{2,i}^{\text{ring}}, d_{3,i}^{\text{ring}}$ | 0.58±0.16 | **0.89±0.05** | 0.48±0.12 |
| $\quad d_{1,i}^{\text{par}}, d_{2,i}^{\text{par}}, d_{3,i}^{\text{par}}$ | **0.63±0.05** | 0.88±0.05 | 0.53±0.11 |
| $\quad$ No SL | 0.58±0.41 | 0.85±0.18 | **0.51±0.36** |
| Washout $\quad d_{1,i}^{\text{par}}, d_{2,i}^{\text{par}}, d_{3,i}^{\text{par}}, d_{2,i}^{\text{wash}}, d_{3,i}^{\text{wash}}, d_{2,i}^{\text{ring}}, d_{3,i}^{\text{ring}}$ | 0.28±0.09 | 0.46±0.09 | 0.22±0.18 |
| $\quad d_{1,i}^{\text{par}}, d_{2,i}^{\text{par}}, d_{3,i}^{\text{par}}, d_{2,i}^{\text{wash}}, d_{3,i}^{\text{wash}}$ | **0.44±0.07** | 0.47±0.10 | **0.33±0.00** |
| $\quad d_{1,i}^{\text{par}}, d_{2,i}^{\text{par}}, d_{3,i}^{\text{par}}, d_{2,i}^{\text{ring}}, d_{3,i}^{\text{ring}}$ | 0.17±0.09 | 0.34±0.02 | 0.00±0.00 |
| $\quad d_{1,i}^{\text{par}}, d_{2,i}^{\text{par}}, d_{3,i}^{\text{par}}$ | 0.33±0.14 | 0.46±0.16 | 0.19±0.27 |
| $\quad$ No SL | 0.41±0.27 | **0.56±0.18** | 0.30±0.22 |
| Capsule $\quad d_{1,i}^{\text{par}}, d_{2,i}^{\text{par}}, d_{3,i}^{\text{par}}, d_{2,i}^{\text{wash}}, d_{3,i}^{\text{wash}}, d_{2,i}^{\text{ring}}, d_{3,i}^{\text{ring}}$ | 0.72±0.05 | 0.78±0.01 | **0.68±0.09** |
| $\quad d_{1,i}^{\text{par}}, d_{2,i}^{\text{par}}, d_{3,i}^{\text{par}}, d_{2,i}^{\text{wash}}, d_{3,i}^{\text{wash}}$ | 0.67±0.06 | 0.77±0.06 | 0.54±0.10 |
| $\quad d_{1,i}^{\text{par}}, d_{2,i}^{\text{par}}, d_{3,i}^{\text{par}}, d_{2,i}^{\text{ring}}, d_{3,i}^{\text{ring}}$ | **0.78±0.20** | **0.87±0.11** | 0.68±0.19 |
| $\quad d_{1,i}^{\text{par}}, d_{2,i}^{\text{par}}, d_{3,i}^{\text{par}}$ | 0.64±0.05 | 0.64±0.08 | 0.61±0.00 |
| $\quad$ No SL | 0.65±0.13 | 0.67±0.15 | 0.54±0.10 |

Best per concept in **bold**. *Abbreviations:* AUC = area under the ROC curve; SL = Soft labels, $N_l$ = Number of lesions, HCC = Hepatocellular Carcinoma. APHE performance cannot be reported on the test set because it contains only APHE-positive lesions.

## Appendix J. Performance Evaluation with Interventions

Table 4 reports HCC classification performance when we progressively intervene on the predicted concepts. The rows 0, 25, 50, 75 and 100 correspond to the number of concept-level interventions applied on the validation or test set. Each intervention consists in correcting a single predicted concept (APHE, washout or capsule) for one lesion and replacing it with its ground-truth value before recomputing the LR-5 decision.

As the number of corrected concepts increases, ROC AUC, PR AUC, MCC and F1 steadily improve on both validation and test sets. On the test set, ROC AUC rises from 0.74 with no intervention to 0.96 with 50–100 concept edits, while MCC increases from 0.42 to 0.91 and F1 from 0.84 to 0.97. This shows that many classification errors can be resolved by editing a relatively small number of concept predictions, without retraining the model. Once the Li-RADS–inspired concepts are made accurate through limited human intervention, the concept bottleneck becomes highly informative for HCC classification. This experiment therefore illustrates the potential of our framework for interactive use: radiologists can correct a few key concepts on ambiguous cases and obtain substantially improved LR-5 predictions.

Table 4: Effect of concept-level interventions on HCC classification performance

| Interventions | ROC AUC↑ | PR AUC↑ | MCC↑ | F1↑ |
|---|---|---|---|---|
| **Validation** (3-folds CV; $N_l$=67, 69, 59; $HCC = 44, 55, 44$) | | | | |
| 0 | 0.78±0.04 | 0.90±0.04 | 0.52±0.07 | 0.88±0.04 |
| 25 | 0.86±0.03 | 0.92±0.04 | 0.69±0.02 | 0.92±0.02 |
| 50 | **0.96±0.03** | **0.99±0.01** | 0.84±0.08 | **0.96±0.03** |
| 75 | **0.96±0.03** | **0.99±0.01** | **0.85±0.09** | **0.96±0.03** |
| 100 | **0.96±0.03** | **0.99±0.01** | **0.85±0.09** | **0.96±0.03** |
| **Test** ($N_l = 55$; $HCC = 38$) | | | | |
| 0 | 0.74±0.03 | 0.88±0.02 | 0.42±0.05 | 0.84±0.00 |
| 25 | 0.81±0.02 | 0.90±0.02 | 0.57±0.02 | 0.88±0.00 |
| 50 | **0.96±0.01** | **0.98±0.00** | **0.91±0.00** | **0.97±0.00** |
| 75 | **0.96±0.01** | **0.98±0.00** | **0.91±0.00** | **0.97±0.00** |
| 100 | **0.96±0.01** | **0.98±0.00** | **0.91±0.00** | **0.97±0.00** |

For each fold, the maximum number of possible interventions equals the total number of concept predictions ($3 \times N_l$): 201, 207 and 177 for the three validation folds, and 165 for the test set. Row labels (0, 25, 50, 75, 100) denote the number of concept edits applied. Best per column in **bold**. *Abbreviations:* AUC = area under the ROC curve; $N_l$ = Number of lesions; HCC = Hepatocellular Carcinoma; F1 = F1-score.

Moreover, the model does not reach perfection using our soft implementation of the Li-RADS guidelines. This can be partly explained by lesions for which the model is not very confident: in such cases, the soft Li-RADS implementation yields an LR-5 probability around or below 0.5, while the ground-truth label is 1. However, providing a continuous LR-5 score instead of a hard binary decision based on fixed rules may actually be more informative and interpretable, especially for lesions with weak APHE, washout or capsule signals.

## Appendix K. Example of predictions

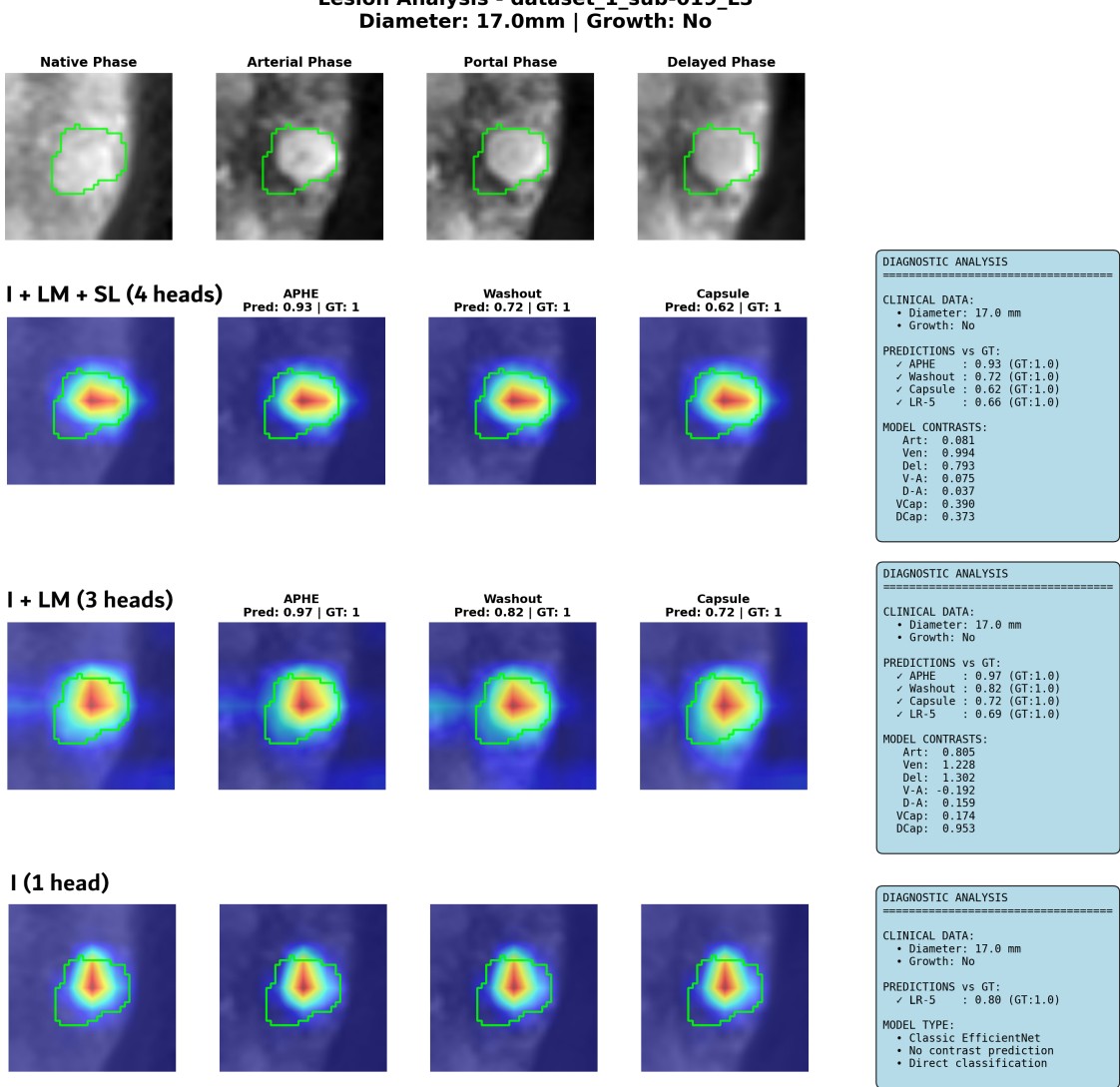

Figure 6: Comparison of predictions from each model.

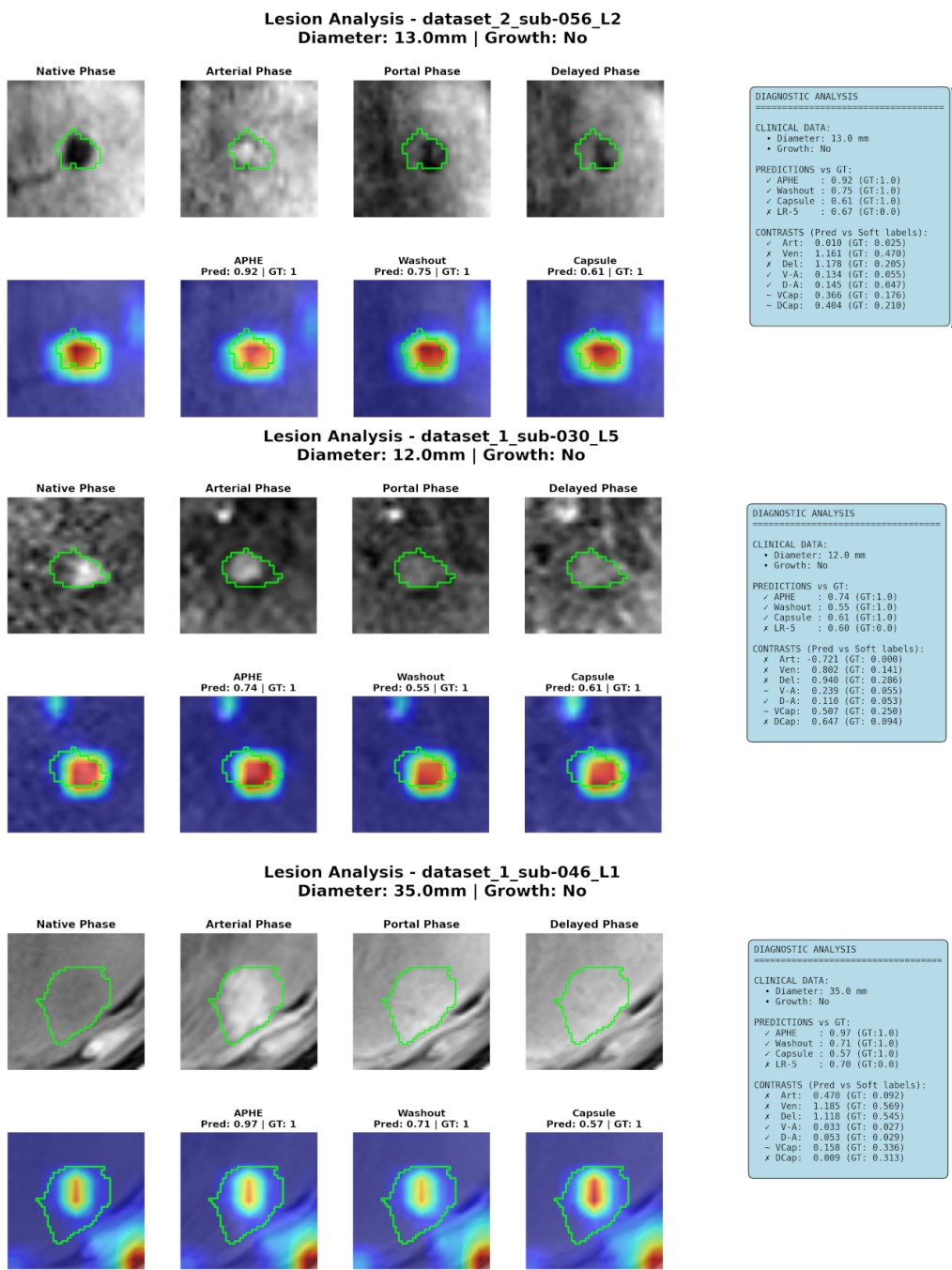

Figure 7: False positive predictions from the I + LM + SL (4 heads) model.

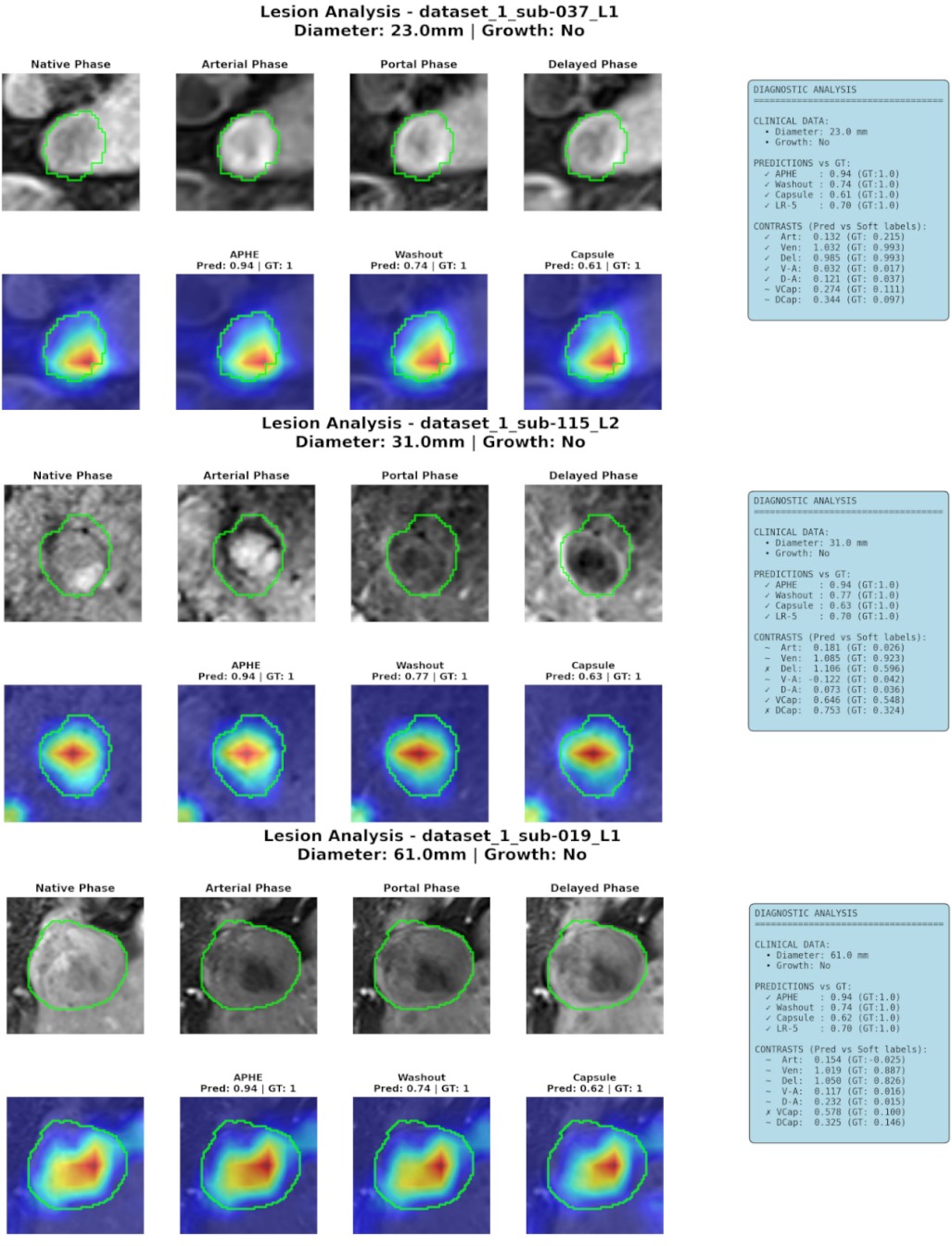

Figure 8: True positive predictions from the I + LM + SL (4 heads) model.

