# OpenReview forum: "Explainable HCC Diagnosis on Dynamic Contrast-Enhanced MRI with a Li-RADS Concept Bottleneck"
_MIDL.io/2026/Conference — MIDL 2026 Poster_

### Official Review · Reviewer_BRPP · 2025-12-19

**Confidence:** 5
**Preliminary Rating:** 2
**Final Rating:** 3

**Summary:**

This paper presents an end-to-end Concept Bottleneck Model (CBM) designed for the diagnosis of Hepatocellular Carcinoma (HCC) using Dynamic Contrast-Enhanced (DCE) MRI. The authors introduce a "physics-driven" approach by predicting intermediate soft labels based on intensity differences (e.g., arterial phase hyperenhancement, washout) and mapping them to Li-RADS concepts. The final classification of LR-5 (definite HCC) is achieved not through a learned classifier, but via a differentiable implementation of the Li-RADS v2018 logical guidelines. The method is evaluated on a dataset of 250 lesions, focusing on the trade-off between diagnostic performance and explainability metrics such as Grad-CAM stability and faithfulness.

**Strengths:**

Clear motivation; elegant mapping of clinical guidelines to differentiable logic.

Detailed mathematical appendices.

The differentiable implementation of the specific Li-RADS v2018 table is a clever engineering contribution.

The intervention analysis (Table 3) is compelling.

**Weaknesses:**

The narrative downplays the significant drop in diagnostic performance (Test MCC drops from 0.55 in the baseline to 0.42 in the proposed method).

Dependence on manual/perfect segmentation for the "soft label" generation is a hidden constraint on scalability.

The conceptual novelty is incremental, building on existing CBM frameworks (Koh et al., 2020). The "physics-driven" aspect is essentially hard-coding radiological heuristics (median intensity differences) into the loss function, which limits the network's ability to learn non-linear texture features that might be diagnostic but not captured by median intensity.

The primary model fails to beat the baseline on accuracy. The dataset is small and single-institution (merged cohorts).

**Detailed Comments:**

I have no minor comments.

**Justification Of Final Rating:**

initial justification of final grade:
I gave reject, this time is strong reject  just to clarify that the paper has some really problems to fix before considering publication.
The core diagnostic utility is sacrificed without providing a proven clinical benefit.  The significant drop in MCC (0.55 to 0.42) and AUC (0.81 to 0.74) is not merely a "trade-off" but a failure to meet the baseline requirements for a high-stakes clinical tool; an interpretable model that is substantially less accurate risks providing "clear explanations" for incorrect diagnoses, which is potentially more dangerous than a black-box model. Furthermore, the authors admit that their "physics-driven" soft labels—the primary technical contribution—actually introduce negative transfer for critical features like the "Capsule" and "Washout."

Updated score on final note:
there were some points clarified by the authors.

**Justification Of The Preliminary Rating:**

While the methodology is rigorous and the "human-in-the-loop" intervention capability is promising, the significant drop in diagnostic performance (AUC 0.81 to 0.74) compared to a standard baseline is a critical failure for a clinical AI paper. A diagnostic tool that is "interpretable" but misses significantly more cancers (lower sensitivity/MCC) is difficult to justify. The reliance on heuristic "median intensity" features likely over-constrains the model.

**Questions To Address In The Rebuttal:**

Weaknesses section is self-contained about questions, please answer them.

Further:
1-The proposed 4-head model sees a substantial drop in MCC (0.55 to 0.42) compared to the baseline. Does this suggest that the "physics-driven" constraints (median intensities) are overly rigid and discarding discriminative texture features found in the pixel data?

2-Segmentation Dependency: Your soft labels require specific ROI definitions (dilated rings, parenchyma). How does the model performance degrade if these masks are generated by an automated segmentation tool rather than manual annotation?

3-The Li-RADS logic assumes APHE, Washout, and Capsule are distinct. However, your results show that adding soft-label supervision hurts Capsule prediction significantly (MCC 0.56 to  0.32). Does this imply that "Capsule" appearance in MRI is not well-represented by simple intensity deltas?

4-Given the lower AUC, have you tested if providing these explanations to a radiologist actually improves their accuracy compared to the baseline's output? High "explanation stability" does not necessarily equal clinical utility if the underlying prediction is wrong.

---

> ### Author Response · Authors · 2026-01-25
>
> - The conceptual novelty is incremental
>
> We appreciate this point. We now explicitly frame the work as a LI-RADS–grounded decision-support system that builds on existing CBM + test-time intervention ideas (Koh et al., 2020). The “physics-driven” intensity deltas are auxiliary soft-label targets to encourage alignment with enhancement dynamics, while the network still learns from the full multi-phase crop (not a hard-coded constraint). Importantly, we added an ablation showing these surrogates are concept-dependent—helpful for APHE but not for washout/capsule—supporting the limitation that imperfect surrogates can introduce noise/negative transfer. (Intro p.3; Results p.10; App. I p.26; Conclusion p.12)
>
> - The primary model fails to beat the baseline on accuracy. The dataset is small and single-institution (merged cohorts).
>
> We agree. We now state explicitly that the full 4-head model (I+LM+SL) does not outperform the single-head baseline on LR-5 test accuracy, and we present the models as different points on an accuracy–interpretability frontier: I+LM preserves LR-5 performance while improving explanation robustness, whereas I+LM+SL can improve robustness but at a non-trivial LR-5 cost. We also added a small external sanity check (HCCSeg; Nl=10) and frame it as preliminary due to limited size. (Table 1 p.9; Sec. 2.1 p.3–4; Table 2 p.11; Conclusion p.12)
>
> 1. The reviewer suggests an interesting explanation for the drop. We agree the MCC drop (0.55to 0.42) reflects a non-trivial accuracy–interpretability trade-off for the 4-head setting, but we do not interpret it as the model being forced to ignore texture: the image backbone still learns from the full pixel data, while the “physics-driven” terms are only auxiliary soft-label regularizers on concept heads. Our new ablation (Table 3) indicates the effect is concept-dependent—helpful for APHE, but neutral/detrimental for washout/capsule—consistent with imperfect surrogate targets and multi-task negative transfer rather than overly rigid constraints. We therefore present the contrast-surrogate supervision as an optional, concept-specific regularizer and clarify this trade-off in the Results/Conclusion. (Table 1 p.9; Appendix I/Table 3 p.26; Conclusion p.12)
>
> 2. Thank you for raising this point. Our ROI-based soft labels require liver and lesion masks to define parenchyma and ring ROIs, and we now state this explicitly. The liver mask is generated automatically with a pre-trained nnU-Net (as in our prior work [1]) , and the lesion mask is provided as an input channel. We did not quantify the impact of replacing expert lesion masks with automated masks; we add this as a limitation and note that major segmentation failures could corrupt the surrogate targets. Future work will evaluate concept/LR-5 performance using automated masks. (Sec. 2.4 p.4; App. A p.16; App. B–C p.17–18).
>
> 3. Yes, we performed an ablation experiment and figured out that our surrogate for capsule decrease performance. We stated clearly that capsule is a challenging concept and that our current soft label (peri-lesional rim enhancement via ring deltas) is a coarse approximation. We revised the discussion to reflect that capsule may require additional cues (morphology/texture/edge patterns, or 3D continuity) beyond median intensity differences. See modification in Results (Table 2) – page 10-11 and Conclusion page 12.
>
> 4. We thank the reviewer for this important point. We did not conduct a reader/user study in this submission, and we do not claim that improved explanation stability necessarily improves clinical decision-making—especially when the prediction is wrong. Our goal is a second-reader decision-support setting: the explanation panel exposes radiologist-relevant intermediate cues to support verification rather than blind trust. We have previously evaluated AI decision-support with radiologists in a clinical workflow setting for aneurysm detection [2]. In the present work, we therefore position a dedicated multi-reader study (baseline output vs. baseline + our panel; accuracy, time-to-decision, confidence calibration, inter-/intra-reader reproducibility) as the key next step; our intervention analysis only provides a preliminary proxy for human-in-the-loop interaction. (See Conclusion, p.12.)
>
> [1] Monnin K, et al. (2025). Deep learning for automatic detection of hepatocellular carcinoma in dynamic contrast-enhanced MRI. Abdominal Radiology. 1-14. 10.1007/s00261-025-05249-4.
>
> [2] Di Noto et al. (2025). Assessing workflow impact and clinical utility of AI-assisted brain aneurysm detection: a multi-reader study. 10.48550/arXiv.2503.17786.

---

> > ### Comment · Reviewer_BRPP · 2026-02-01
> > **not good enough for MIDL**
> >
> > I truly appreciate authors' rebuttal.
> > However, despite the authors’ efforts to frame their model as an "accuracy-interpretability frontier" choice, the paper remains a reject to my opinion because the rebuttal confirms that the core diagnostic utility is sacrificed without providing a proven clinical benefit.
> >
> > The significant drop in MCC (0.55 to 0.42) and AUC (0.81 to 0.74) is not merely a "trade-off" but a failure to meet the baseline requirements for a high-stakes clinical tool; an interpretable model that is substantially less accurate risks providing "clear explanations" for incorrect diagnoses, which is potentially more dangerous than a black-box model.
> >
> > Furthermore, the authors admit that their "physics-driven" soft labels—the primary technical contribution—actually introduce negative transfer for critical features like the "Capsule" and "Washout."
> >
> > By positioning a reader study and automated segmentation validation as "future work," the authors leave the paper’s practical value entirely theoretical, failing to demonstrate that the improved explanation stability actually leads to better human-AI team performance to compensate for the underlying loss in sensitivity.

---

> > > ### Author Response · Authors · 2026-02-02
> > > **Clarification on the revised claims and evidence**
> > >
> > > Thank you for the follow-up. We respectfully disagree with the characterization that our contribution is “entirely theoretical” because a reader study and automated-mask validation are listed as next steps. A reader study is the final step to quantify downstream human performance, but it is not a prerequisite to establish the practical value of a decision-support method at this stage. The revised paper already provides task-level evidence that the interface is actionable: concept-level corrections propagate to the final LR-5 output and can substantially improve the decision without retraining. This directly tests the intended human-in-the-loop mechanism (verification and correction), beyond reporting stability metrics alone.
> > >
> > > Importantly, the claim that we “sacrifice core diagnostic utility” does not reflect the revised framing. We no longer present the 4-head (I+LM+SL) as the recommended operating point; it is explicitly treated as an ablation/stress test of contrast-surrogate supervision, and we transparently report that it can incur a non-trivial LR-5 cost and negative transfer for washout/capsule. The recommended practical variant in the revision is the 3-head I+LM, which maintains LR-5 discrimination at a level comparable to the single-head baseline while providing substantially improved robustness/interpretability signals. Thus, we are not asking clinicians to accept a materially worse diagnostic model “in exchange for explanations.”
> > >
> > > Regarding the “physics-driven” soft labels, the revised paper does not claim they universally help. The ablations show the effect is concept-dependent: APHE benefits from the soft labels, whereas washout/capsule can degrade. This is precisely why we position these targets as soft auxiliary supervision rather than hard constraints: LI-RADS feature assessment, especially for washout/capsule, is known to be variable across annotators, so a continuous, uncertainty-aware surrogate can be appropriate for some concepts but should not be treated as a one-size-fits-all improvement.
> > >
> > > Finally, the argument that explanations are “more dangerous” if the prediction is wrong is exactly why our design is not a single post-hoc heatmap attached to a black box. We provide radiologist-relevant intermediate cues and an explicit mechanism to disagree and correct at the concept level, which is not possible with a standard single-score baseline. In short: the revised paper makes clear what is recommended (I+LM), what is exploratory (I+LM+SL), what improves (APHE and intervention-driven correction), and what remains the appropriate next step (formal multi-reader evaluation).

---

> > > > ### Comment · Reviewer_BRPP · 2026-02-02
> > > > **you have a point, thank you for clarifying!**
> > > >
> > > > Thank you for the clarification!
> > > > I read it carefully.

---

### Official Review · Reviewer_aPoE · 2026-01-09

**Confidence:** 4
**Preliminary Rating:** 3
**Final Rating:** 4

**Summary:**

This paper proposes an explainable deep learning framework for hepatocellular carcinoma diagnosis based on the LI-RADS. The method adopts a concept-bottleneck-style architecture, predicting imaging concepts via multiple task heads and combining them through a differentiable approximation of LI-RADS decision rules. The authors emphasize explanation quality, stability, and clinical alignment, and evaluate the framework on a multi-site MRI dataset.

**Strengths:**

- The problem formulation is aligned with real clinical workflows.
- The bottleneck architecture with soft logical operators for LI-RADS reasoning is intuitive.
- The use of uncertainty-weighted multi-task learning, border regularization, and intensity-based soft labels seems not to be an ad hoc design.
- The evaluation is  incorporating multi-center data, cross-validation, multiple baselines, and both diagnostic and explainability metrics.

**Weaknesses:**

- Qualitative visualization is limited despite the paper’s emphasis on explainability.
- Statistical significance testing is absent for the reported improvements.
- Although the method is carefully formulated, the core contribution lies primarily in integrating existing ideas rather than introducing fundamentally new concepts.
- The paper does not clearly discuss when the proposed trade-offs would be acceptable in real clinical practice.
- The explanation metrics remain indirect proxies; measures such as explanation “accuracy” and stability are informative but still abstract.

**Detailed Comments:**

- Some model variants show worse diagnostic performance than simpler baselines, which raises the question of whether the improved explainability comes at a non-trivial performance cost. This trade-off should be clarified.
- The paper would benefit from more qualitative analysis, especially representative and failure cases, ideally with radiologist-style annotations explaining why an explanation is helpful or potentially misleading.
- It would help if the authors more clearly distinguished what is truly novel from what is adapted from prior work, possibly reframing the contribution as a clinically grounded system rather than a purely methodological advance.
- A short ablation or discussion on when and why explanation quality may degrade diagnostic performance would improve transparency.
- Including a simple statistical test across lesions would significantly strengthen the empirical claims.

**Justification Of Final Rating:**

The rebuttal substantially strengthens the paper by adding formal statistical testing, expanding qualitative evidence with representative success and failure cases, clarifying the performance–interpretability trade-offs, and more accurately framing the contribution as a clinically grounded decision-support system.

However, evidence for clinical usefulness remains largely indirect and proxy-based rather than outcome-validated. The qualitative analysis, while meaningfully improved, is still limited in scope and illustrative rather than comprehensive. In addition, the contribution is best viewed as a careful and well-integrated system built from existing ideas, rather than a conceptually transformative methodological advance.

**Justification Of The Preliminary Rating:**

This paper addresses an important and timely problem and is thoughtfully engineered with strong clinical motivation. However, the contribution is largely incremental, and the evidence supporting improved explainability is not fully convincing due to limited qualitative analysis and the lack of statistical validation.

**Questions To Address In The Rebuttal:**

Please check the above weakness and Detailed Comments sections.

---

> ### Author Response · Authors · 2026-01-25
>
> •	Statistical significance.
>
> We agree, and we have added statistical significance testing for the reported improvements (EfficientNet vs. DETECT). A a two-sided Wilcoxon signed-rank test to paired values for continuous explanation metrics (e.g., gradient overlap, geometric stability/instability score, and intensity instability measured by MAD), and a McNemar test to paired binary outcomes (e.g., pointing-game hit/miss and thresholded LR-5 pointing decisions). We now report the resulting p-values in the revised manuscript. See modifications in Section 2.7.2 – page 9 and Table 1 – page 9.
>
> •	The explanation metrics remain indirect proxies
>
> We agree, and we revised the manuscript to make this limitation explicit. In the new version, we clarify that explanation “accuracy”/stability are proxy indicators of robustness and plausibility, and do not in themselves establish clinical utility or safety, which ultimately requires clinician-centered evaluation.
> To better ground interpretation, we strengthened the clinical perspective / intended use framing (decision support / second reader rather than autonomous diagnosis) and explicitly discuss acceptable operating points (e.g., sensitivity-first use cases), in line with recent reporting guidance that emphasizes stating purpose and intended clinical decision context.  We also updated the Results to include a small external sanity check (Nl=10; HCC=5) and added cautionary wording that these estimates are imprecise and are reported to probe behavior under domain shift rather than to claim definitive generalization—consistent with guidance on the pitfalls of small external validation samples.
> Finally, we position our concept-edit intervention as a proxy for interactive clinician correction and explicitly describe a radiologist user study as future work to directly assess clinical usefulness. See modifications in Conclusion – page 12.
>
> •	Trade-off clarification
>
> We agree and now make the performance–interpretability trade-off explicit. We clarify the intended use as a second-reader decision-support tool, and present the variants as different points on an accuracy–interpretability frontier: the 3-head (I+LM) offers the best balance in-domain, while 4-head (I+LM+SL) improves robustness but can incur a non-trivial LR-5 cost. This is consistent with known trade-offs in concept-based models where added structure/intervenability can reduce task accuracy when concepts/surrogates are imperfect. We avoid over-claiming diagnostic gains and note that better explanation metrics do not guarantee better diagnosis. (Results p.9; Conclusion p.12)
>
>
> •	Qualitative analysis
>
> We agree that the initial version did not include enough qualitative evidence. In the revised manuscript, we have therefore expanded the qualitative visualization by adding a case-based figure with 6 representative examples from the I + LM + SL model: 3 true positives and 3 false positives, each showing the multi-phase lesion crop with the lesion contour and the corresponding attribution maps for the predicted concepts. This directly complements the quantitative explainability metrics by providing intuitive, sample-level illustrations (including success and failure modes). See modifications in Results – page 11 and Appendix K – page 28-30.
>
>
> •	True novelty
>
> Thank you for the suggestion. We revised the Introduction and Main contributions to frame the work as a clinically grounded LI-RADS decision-support system (LR-5 prediction + radiologist-relevant concepts) rather than a generic methodological XAI advance. We also clarify what is adapted (concept bottlenecks, saliency maps) versus what is new in our setting: DCE/contrast-driven soft labels for major features, a differentiable LI-RADS rule to produce LR-5 probability, and an LI-RADS-specific intervention/robustness evaluation protocol. (Introduction, p.3).
>
> •	Ablation
>
> Thank you for the suggestion. We added an ablation of the contrast-surrogate soft-label supervision (Table 3, Appendix I) showing it improves APHE concept prediction but can be neutral or detrimental for washout/capsule, illustrating that interpretability-oriented supervision can introduce noise or negative transfer depending on concept quality. We also briefly discuss why this can happen (concept-bottleneck information loss, multi-task negative transfer/noisy concepts, and stability–fidelity trade-offs), and emphasize that improved explanation metrics do not necessarily translate into diagnostic gains. (Results p.10; App. I/Table 3 p.26; Conclusion p.12)

---

### Official Review · Reviewer_udKT · 2026-01-09

**Confidence:** 4
**Preliminary Rating:** 4
**Final Rating:** 4

**Summary:**

This paper presents an explainable deep learning approach for diagnosing HCC from multi-phase contrast-enhanced liver MRI that is explicitly aligned with LI-RADS. Instead of directly predicting the final diagnosis, the model first learns clinically meaningful imaging features such as arterial phase hyper-enhancement, washout, and capsule appearance, and then combines them using a differentiable version of the LI-RADS decision rules. The authors show that this concept-based design improves explanation quality, robustness, and interpretability while maintaining competitive diagnostic performance. Overall, the work highlights how embedding clinical reasoning into model design can lead to more trustworthy AI systems for liver imaging.

**Strengths:**

1. This paper is a solid example of integrating clinical knowledge into the computation of LI-RADS 5 probability.
2. The idea of incorporating phase-difference images to derive auxiliary soft labels for hyper-/hypo-enhancement, washout, and peri-lesional rim enhancement is particularly compelling.
3. The paper is mathematically sound.
4. The literature review is adequate and well structured.
5. The ablation study analyzing the trade-off between explainability and classification performance is interesting and informative.

**Weaknesses:**

1. Training stability may be an issue due to the extensive use of sigmoid layers in the computation of the LI-RADS probability.
2. Grad-CAM may not be the most suitable choice for generating explanation maps for this neural network model.
3. There are no external baseline methods included for comparison in LI-RADS 5 probability estimation.

**Detailed Comments:**

Major Comments:
1. For readers outside the liver imaging community, it may be helpful to briefly explain how the native, arterial, portal venous, and delayed phases are acquired. For example, imaging without contrast, imaging immediately after intravenous contrast injection, imaging after a short delay, and imaging some time after contrast administration.
2. How did you identify the shortcut learning behavior? Was it inferred from observing unusual explanation maps generated by Grad-CAM? If so, it is worth noting that Grad-CAM is known to have limited clinical reliability in the explanations it produces. More recent and robust explanation methods, such as NormGrad [1, 2], might provide more trustworthy insights.
3. While the motivation for using the Weighting Game for explainability evaluation is reasonable, given the roughly round shape of benign and malignant liver lesions, the Pointing Game could also be a suitable alternative for cases involving a single lesion.
4. Although the differentiable implementation of LI-RADS appears mathematically sound, the use of sigmoid layers to distinguish between lesion size categories (≥ 20 mm vs. 10–19 mm) may lead to gradient saturation during training. This could negatively affect learning, particularly for very small or very large lesions.
* What is the lesion size distribution in your dataset?
* Since $k$ is used as a temperature hyperparameter, was it tuned using the validation split?
* Have you considered alternatives to sigmoid-based gating with temperature scaling?
5. I am not fully convinced by the use of the uncertainty-based loss weighting scheme, as the AUC performance appears to degrade when additional loss terms are introduced. Did you explore alternative strategies, such as tuning the balance between the main loss and the soft-label losses via random search?
6. Although the introduction discusses related methods that estimate LI-RADS 5 probability from imaging data (e.g., Li et al., 2025), these approaches do not appear among the comparative baselines for LR-5 classification performance. Including alternative CNN backbones for the single-head model, such as ResNet-34, would strengthen the experimental evaluation.

Minor Comments:
1. Tables 1 and 2 would benefit from additional horizontal or vertical lines to improve readability.
2. Please clarify the values used for $C_\{in\}$ and $\lambda_{border}$?

[1] Rebuffi et al. "There and Back Again: Revisiting Backpropagation Saliency Methods" CVPR 2020

[2] Ozer, C., Oksuz, I. "Explainable Image Quality Analysis of Chest X-Rays" MIDL 2021

**Justification Of Final Rating:**

Although the authors state that NormGrad has been adopted in the revised version, the resulting explanation maps appear visually indistinguishable from those previously produced using Grad-CAM. Given the known methodological differences between these approaches, one would expect observable visual and/or quantitative differences; however, no such differences are demonstrated or analyzed. As a result, it remains unclear whether NormGrad was actually applied or whether it provides any additional insight.

Furthermore, while the proposed approach is evaluated on a so-called external split, the distinction between the test set and the External Cohort is not clearly defined, nor are the data acquisition protocol and selection criteria for the External Cohort described. This lack of detail prevents assessment of the true external validity of the results.

These issues significantly limit the credibility of the revision and are the primary reasons why my original assessment remains unchanged. Despite these concerns, I believe the paper could still be suitable for consideration at this year’s MIDL after fixing these issues and careful proofreading.

**Justification Of The Preliminary Rating:**

I like the step-by-step integration of clinical knowledge into the LI-RADS decision-making process. The proposed architecture also predicts the existence of APHE, Washout, and Capsule features, which are used to determine the LI-RADS level. Addressing the concerns outlined above could justify a rating of 5 after the rebuttal phase.

**Questions To Address In The Rebuttal:**

Please check the detailed comments section.

---

> ### Author Response · Authors · 2026-01-25
>
> Major Comments:
> 1. We thank the reviewer for this comment and agree this is helpful for readers outside liver MRI. We added a short explanation in the main text. See modifications in Section 2.3 - page 4, Appendix A - page 16.
>
> 2. Thank you for this suggestion, we agree that CAM-style heatmaps can be unreliable if interpreted uncritically, and we now explicitly acknowledge this limitation and avoid relying on Grad-CAM for the revised analysis, consistent with broader evidence that saliency maps can be visually plausible yet misleading. Following the reviewer’s suggestion, we have replaced Grad-CAM with NormGrad in our pipeline and recomputed the reported explanation results/metrics accordingly, updating the manuscript to clearly state the method used. See modifications in Section 2.4 – page 4, Appendix B – page 17 and Section 2.7.2 – page 9
>
> 3. We thank the reviewer for this suggestion. In the revised manuscript, we now report the Pointing Game as a complementary explainability metric. See modifications in Section 2.7.2 – page 9 and Table 1 – page 9.
>
> 4. Thank you for this comment. We would like to clarify a potential misunderstanding: we do not introduce an additional sigmoid layer in the network. The sigmoid is used only as a fixed gating function to smoothly approximate the hard LI-RADS size thresholds, not as a learned network activation. Since lesion diameter is a measured scalar from the segmentation (not a learned feature), saturation does not impede backpropagation through the image encoder; it simply yields a near-binary branch selection far from the threshold, while the temperature controls the smooth transition near 10–19 mm vs ≥20 mm. (Sec. 2.5.3, p.6; App. E, p.21)
>
> - We agree that lesion size is important context. We clarified the lesion size distribution in the revised manuscript.
>
> - We did not treat k as a separately tuned hyperparameter in the main study: we fixed k=3 based on pilot runs on the validation split and kept it constant across all model variants for a fair comparison (temperature-like parameters are typically selected on a held-out validation set). As an additional sanity check, when we feed ground-truth APHE/washout/capsule into the LR-5 rule head, it reaches near-ceiling performance—indicating the differentiable LR-5 head itself is not the bottleneck. See modifications in Appendix E – page 21.
>
> - Yes—but we clarified that this is not a learnable sigmoid layer inside the network. The “sigmoid gate” is a fixed, temperature-controlled smooth approximation of the LI-RADS size threshold (i.e., a differentiable surrogate of a step/indicator function), applied to a non-learned scalar lesion diameter. Under this interpretation, the relevant alternatives are essentially other smooth step functions, which would be largely interchangeable in our use case (they mainly differ in how sharply and where they transition).
>
>
> 5.We did not perform a random search over loss-weight combinations. Instead, we used uncertainty-based loss weighting (Kendall et al.) as a principled way to reduce manual tuning and improve multi-task stability. In our data, APHE positives are substantially more frequent than washout/capsule, and with fixed weights we observed APHE tending to dominate the shared representation; uncertainty weighting yielded more balanced and consistent concept predictions (APHE/washout/capsule), which is important because the differentiable LI-RADS rule relies on these intermediate estimates. See modifications in Section 2.5.4 – page 7 and Appendix F – page 22.
>
> 6. We agree that comparisons to prior LI-RADS–related AI work are important, but directly adding methods such as Li et al. (2025) as LR-5 probability baselines is not an apples-to-apples comparison in our setting. Li et al. focus on HCC vs non-HCC classification on gadoxetic-acid MRI using multi-sequence inputs with manual bounding boxes, and infer LI-RADS features post hoc from classifier activations, whereas we train an end-to-end concept bottleneck and compute LR-5 probability via a differentiable LI-RADS rule under our input protocol. Moreover, key LI-RADS feature definitions depend on the contrast agent (e.g., washout assessment differs for gadoxetate vs extracellular agents), making cross-study numeric comparisons sensitive to protocol and labeling. Therefore, we prioritize a controlled comparison against a strong single-head baseline trained/evaluated on identical data and preprocessing to isolate the effect of the concept-based design and LI-RADS reasoning; backbone swaps (e.g., ResNet-34) are orthogonal and can be explored in follow-up ablations.
>
> Minor Comments:
> 1. Agreed; we updated formatting for readability. See modifications in Table 1 – page 9 and Table 2 – page 11.
> 2. Thank you for pointing this out. We used C_in=5×5=25 input channels and we used a fixed coefficient λ_border=0.1 in all experiments. We have updated the in Section 2.4 – page 4-5, Section 2.6.1 – page 7 and Appendix C – page 18.

---

> > ### Comment · Reviewer_udKT · 2026-01-25
> > **Reference for NormGrad**
> >
> > I think the reference you provided for NormGrad "(Chen et al., 2017)" is inappropriate.

---

> > > ### Author Response · Authors · 2026-01-26
> > >
> > > Thank you for spotting this. We mistakenly cited Chen et al. (2017) for NormGrad but we intended to cite Rebuffi et al., "There and Back Again: Revisiting Backpropagation Saliency Methods", CVPR 2020. We will correct this reference in the final version.

---

### Author Rebuttal · Authors · 2026-01-25

**Rebuttal:**

Dear Reviewers, thank you for your feedback. We updated the paper with additional analyses/ablations, expanded qualitative failure cases, clearer clinical positioning, and an external evaluation on a new dataset to assess robustness.

**Supporting Material:**

/attachment/b7137f1242818136750e10b924235ce4d6486b30.pdf

---

### Comment · Area_Chair_rdAZ · 2026-01-30
**reminder for the deadline to evaluate authors's responses and provide final scores**

Dear Reviewers,

The authors have submitted responses to all reviewers, along with an updated manuscript in PDF format. We kindly ask reviewers to:

1. Evaluate the authors’ responses and the revised manuscript;
2. Participate in discussions with the authors during the discussion phase;
3. Update the final rating by clicking “Edit” → “Official Review” and providing the Final Rating by 02/01/2026.
Your efforts are extremely helpful in maintaining the high academic quality of MIDL 2026 and in supporting the Area Chairs and Program Chairs in making final decisions. We are getting very close to the deadline, so please check you author's responses.

Thank you very much for your time and contributions!

---

### Meta-Review · Area_Chair_rdAZ · 2026-02-08

**Recommendation:** Accept (Poster)
**Confidence:** 5

**Metareview:**

This paper presents a clinically grounded framework for HCC diagnosis from multi-phase contrast-enhanced liver MRI that explicitly embeds LI-RADS reasoning into model design. Reviewers consistently recognized the strong alignment with clinical workflow, the clarity of the conceptual bottleneck architecture, and the careful formulation of a differentiable LI-RADS decision rule. The technical presentation is sound, well documented, and supported by extensive methodological detail.

The main concerns raised in the initial reviews centered on trade-offs between interpretability and diagnostic performance, limited qualitative and statistical evidence for explainability claims, reliance on expert segmentations, and the incremental nature of the methodological novelty. Reviewers also asked for clearer framing of the intended clinical use case and for stronger justification of the evaluation choices.

The authors responded constructively and thoroughly. In revision, they clarified the intended operating point of the system, explicitly positioning the three-head model as the recommended practical variant that preserves diagnostic performance while improving robustness and interpretability, and reframing the four-head model as an ablation illustrating concept-dependent trade-offs. They replaced Grad-CAM with NormGrad, expanded qualitative examples and failure cases, added complementary explainability metrics, corrected references, and introduced appropriate statistical significance testing. Limitations related to external validation, segmentation dependence, and the proxy nature of explainability metrics are now clearly acknowledged, and the contribution is more accurately framed as a clinically grounded decision-support system rather than a purely methodological advance.

These revisions directly addressed the core reviewer concerns and led to increased confidence and improved scores across reviewers. While the contribution is incremental and the evidence for downstream clinical benefit remains indirect, the work is careful, transparent, and well aligned with real clinical reasoning. I therefore recommend acceptance.

---

### Decision · Program_Chairs · 2026-02-14

Accept (Poster)